# Rapid single-tier serodiagnosis of Lyme disease

Rajesh Ghosh [1,10], Hyou-Arm Joung [2,3,10], Artem Goncharov[2,3], Barath Palanisamy[1], Kevin Ngo[1], Katarina Pejcinovic [1], Nicole Krockenberger [1], Elizabeth J. Horn[4], Omai B. Garner[5], Ezdehar Ghazal[6], Andrew O'Kula[6], Paul M. Arnaboldi[6,7], Raymond J. Dattwyler [6,7], Aydogan Ozcan [1,2,3,8] ✉ & Dino Di Carlo [1,3,9] ✉

Point-of-care serological and direct antigen testing offers actionable insights for diagnosing challenging illnesses, empowering distributed health systems. Here, we report a POC-compatible serologic test for Lyme disease (LD), leveraging synthetic peptides specific to LD antibodies and a paper-based platform for rapid, and cost-effective diagnosis. Antigenic epitopes conserved across *Borrelia burgdorferi* genospecies, targeted by IgG and IgM antibodies, are selected to develop a multiplexed panel for detection of LD antibodies from patient sera. Multiple peptide epitopes, when combined synergistically with a machine learning-based diagnostic model achieve high sensitivity without sacrificing specificity. Blinded validation with 15 LD-positive and 15 negative samples shows 95.5% sensitivity and 100% specificity. Blind testing with the CDC's LD repository samples confirms the test accuracy, matching lab-based two-tier results, correctly differentiating between LD and look-alike diseases. This LD diagnostic test could potentially replace the cumbersome two-tier testing, improving diagnosis and enabling earlier treatment while facilitating immune monitoring and surveillance.

With the increased prevalence of emerging infections and vector-borne illnesses, it is critical to deploy robust and reliable testing platforms to combat the emergence and transmission of diseases[1]. Platforms that can be deployed rapidly and be used in point-of-care (POC) settings or for at-home testing can play a leading role in the rapid deployment of treatments for these diseases[2]. For example, during the COVID-19 pandemic, cost-effective rapid antigen tests and molecular diagnostic tests enabled quick isolation and therapeutic intervention for patients infected with the SARS-CoV-2 virus[3,4]. Lyme disease (LD) is a zoonotic infection caused by spirochetes of the *Borrelia burgdorferi sensu lato* complex that are transmitted through the bite of *Ixodes*

ticks[5]. It is the most prevalent vector-borne disease in North America and Europe[6] (Fig. 1a). The incidence of the disease has continued to rise, exacerbated by climate change and the growing geographic distribution of tick populations[7] (Fig. 1b). Early LD diagnosis typically relies on clinical evaluation, essentially the presence of erythema migrans (EM) skin lesions. However, laboratory confirmation is essential in cases with atypical lesions or extracutaneous signs or symptoms, as these symptoms often overlap with those of other illnesses[8,9]. Early detection and treatment are crucial to prevent the dissemination of bacteria to a variety of distal sites, resulting in serious tissue-specific manifestations, including neurological, cardiac, or

[1]Bioengineering Department, University of California, Los Angeles, CA 90095, USA. [2]Electrical & Computer Engineering Department, University of California, Los Angeles, CA 90095, USA. [3]California NanoSystems Institute (CNSI), University of California, Los Angeles, CA 90095, USA. [4]Lyme Disease Biobank, Portland, Oregon, OR 97221, USA. [5]Department of Pathology and Laboratory Medicine, University of California, Los Angeles, CA 90095, USA. [6]Department of Pathology, Microbiology, and Immunology, New York Medical College, Valhalla, NY 10595, USA. [7]Biopeptides, Corp, Ridgefield, CT 06877, USA. [8]Department of Surgery, University of California, Los Angeles, CA 90095, USA. [9]Department of Mechanical Engineering, University of California, Los Angeles, CA 90095, USA. [10]These authors contributed equally: Rajesh Ghosh, Hyou-Arm Joung. ✉e-mail: ozcan@ucla.edu; dicarlo@ucla.edu

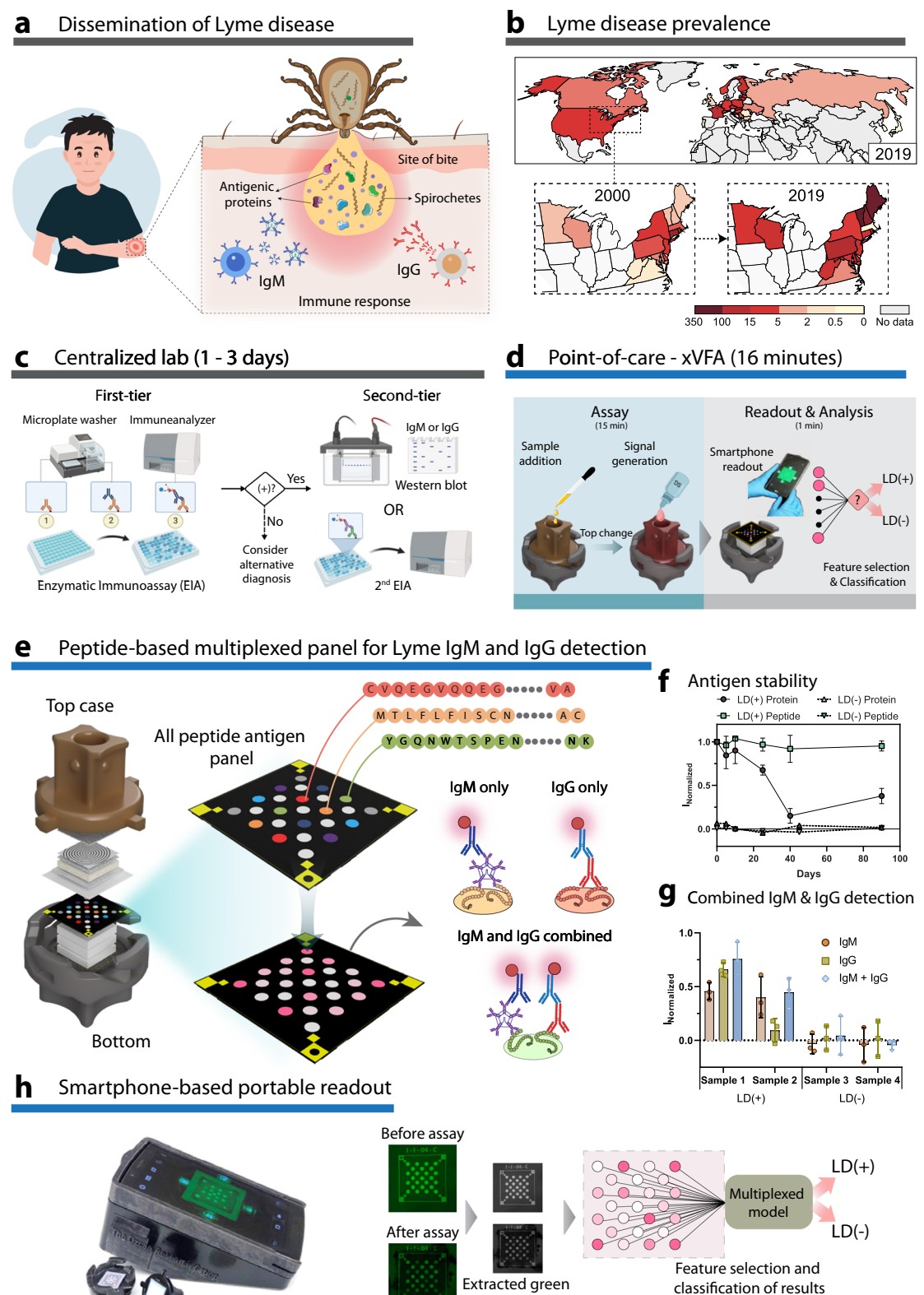

**a** Dissemination of Lyme disease

**b** Lyme disease prevalence

350 100 15 5 2 0.5 0 No data

**c** Centralized lab (1 - 3 days)

First-tier

Microplate washer   Immuneanalyzer

Enzymatic Immunoassay (EIA)

Second-tier

IgM or IgG

Western blot

OR

2$^{nd}$ EIA

Consider alternative diagnosis

**d** Point-of-care - xVFA (16 minutes)

Assay (15 min)

Sample addition   Signal generation

Top change

Readout & Analysis (1 min)

Smartphone readout

LD(+)

LD(-)

Feature selection & Classification

**e** Peptide-based multiplexed panel for Lyme IgM and IgG detection

Top case

All peptide antigen panel

C V Q E G V Q Q E G •••••• V A
M T L F L F I S C N •••••• A C
Y G Q N W T S P E N •••••• N K

IgM only   IgG only

IgM and IgG combined

Bottom

**f** Antigen stability

LD(+) Protein   LD(-) Protein
LD(+) Peptide   LD(-) Peptide

**g** Combined IgM & IgG detection

IgM   IgG   IgM + IgG

Sample 1   Sample 2   Sample 3   Sample 4
LD(+)                  LD(-)

**h** Smartphone-based portable readout

Before assay

After assay

Extracted green channel

Multiplexed model

LD(+)

LD(-)

Feature selection and classification of results

rheumatoid complications[10]. The current standard for laboratory diagnosis is a two-tier testing process conducted in centralized facilities, primarily due to the lack of tests with sufficient specificity for a single-tier approach. With over 3.4 million LD tests conducted each year[11], there is a critical need for a single-tier test that can facilitate rapid diagnosis and treatment[12].

In the absence of EM, serological (antibody) testing of LD remains the mainstay approach for laboratory confirmation[13]. The US Centers for Disease Control and Prevention (CDC) recommends a two-tiered approach for the diagnosis of LD, consisting of a first-tier enzyme immunoassay (EIA) which if positive or equivocal is followed by a second-tier western blot (WB) or a different EIA (Fig. 1c). The two-tier

**Fig. 1 | Overview of the paper-based multiplexed vertical flow assay (xVFA) and point-of-care diagnosis of Lyme disease (LD). a** Transmission of *Borrelia burgdorferi* through the bite of *Ixodes* ticks and the presentation of various antigens generating an immune response from the host. **b** Comparison of incidence of LD cases in the northeastern US from 2000 and 2019 indicating an increase in the incidence of cases due to the growing population of ticks. Worldwide incidence of LD in 2019. Legend indicates the number of cases per 100,000 people. **c** Centralized laboratory-based two-tier serology of LD uses relatively expensive instruments and trained personnel, resulting in high turnaround time and cost per test. **d** Point-of-care xVFA assay using low-cost paper layers and a smartphone reader that provides results for a multiplexed LD assay in <20 min. **e** The xVFA contains a selected peptide panel immobilized on a nitrocellulose membrane that reacts with IgM and IgG antibodies from LD patient serum. **f** Stability of modVlsE-FlaB peptide and VlsE recombinant protein indicating a loss in performance of a protein immobilized assay by more than 50% over a 90-day period. Data were presented as mean values ± SD, with standard deviation indicating three replicates ($N = 3$). **g** Combining IgM and IgG detection in a single xVFA assay enhances the sensitivity of an individual immunoreaction spot ($N = 3$). Data were presented as mean values ± SD. **h** Smartphone-based portable reader and automated image processing of the signals from the peptide panel before and after the assay, yielding normalized signal intensities. The individual peptide spots are analyzed using a multiplexed model to classify samples as either LD positive or negative. Panels **a**, **c**, **d** were created with BioRender.com released under a Creative Commons Attribution-NonCommercial-NoDerivs 4.0 International license (https://creativecommons.org/licenses/by-nc-nd/4.0/deed.en).

system is widely understood to have significant drawbacks, requiring longer turnaround times, underreporting of cases, and a general failure to detect and treat LD in its early stages when treatment is most efficacious at preventing disseminated disease sequelae[14–16]. Furthermore, WB interpretation is subjective and the requirement of multiple specific protein bands to be positive results in failure in detecting most early-stage infections[17]. Additionally, commercial EIAs that are currently available use (i) whole-cell lysates or recombinant proteins from single isolates of B31 species, which have cross-reactive epitopes that are common to other bacteria resulting in a high rate of false positivity[18,19] or (ii) single epitope-based detection, which precludes the recognition of antibodies to other immunodominant epitopes and does not take into consideration the variations in antibody production to different antigens over the time course of infection[20]. Further, these tests are limited in their ability to detect biomarkers against multiple key antigens simultaneously or lack the flexibility to incorporate clinically validated next-generation biomarkers as potential diagnostic targets. Attempts to directly detect the pathogen using culture or molecular techniques have failed due to the transient presence of the bacteria in the bloodstream and very low copy numbers of pathogen nucleic acids[21].

Here, we report a synthetic peptide-based multiplexed vertical flow assay (xVFA) for single-tier POC-compatible diagnosis of LD, overcoming limitations posed by current two-tier tests and traditional protein-based assays. The multiantigen panel consists of synthetic peptide-based immunogenic targets that detect combined IgM and IgG antibody responses from patient serum samples in a single assay, resulting in efficient diagnosis and improved patient outcomes. We use a machine learning-based diagnostic model to interpret the multiplexed results into a diagnostic recommendation yielding 95.5% sensitivity and 100% specificity in a separate validation cohort from the Lyme Disease Biobank, comprising an equal fraction of positive and negative samples. Further, the xVFA matches the performance of standard two-tiered testing in a separate cohort from the CDC, all within a single test. This peptide-based xVFA involves simple operational steps compatible with resource-limited settings such as rural tick-endemic regions. The multiplexed assay requires only 20 μL of serum sample and provides results in <20 min, limiting reagent and sample consumption and drastically improving the turnaround times for LD tests compared to currently available assays in widespread use. This work demonstrates the possibility of replacing traditional lab-based assays with robust multiplexed POC diagnostic platforms, promoting distributed healthcare systems and increased disease surveillance in the community. This is particularly relevant in the current public health landscape, with the COVID-19 pandemic highlighting the need for effective distributed diagnostic tools.

## Results
### Designing a single-tier POC-compatible assay for serologic testing of Lyme disease
The xVFA platform leverages a multiplexed array of immunoreactive peptide epitopes from *B. burgdorferi* and the ease of use of low-cost

paper-based sensors to provide a single-tier, rapid, and accessible platform to test for LD. Figure 1e–g illustrates an overview of the paper-based xVFA platform, which consists of multiple layers of paper with tuned flow properties that are stacked vertically. The different paper layers are assembled to ensure a uniform flow of samples and assay fluids across the entire cross-section of the sensing region, yielding an independent but relatively uniform environment for convection and reaction at each of an array of peptide spots. We were able to multiplex up to 25 immunoreaction spots with less than 8% CV and utilized a total of nine different peptide antigens deposited in duplicates for all reported results unless otherwise stated (Fig. 1e). In a previous study, we developed a two-tier assay using a combination of recombinant proteins and a single peptide that detected individual IgM and IgG levels in serum for detection of LD[19]. However, we observed that recombinant proteins suffer from poor specificity as they could potentially cross-react with other patient samples, particularly look-alike diseases or other common bacterial infections, leading to false-positive results in Lyme disease diagnosis[18]. Apart from potentially low cross-reactivity and high specificity, synthetic peptides have additional advantages of reduced cost and increased shelf-life when compared to full-length recombinant proteins of *B. burgdorferi* (Fig. 1f), which are critical features in a POC test. The modVlsE-FlaB peptide on the xVFA maintained >95% reactivity over 90 days stored at room temperature, while full-length recombinant protein lost more than half of its reactivity under the same storage conditions and time. Additionally, the combined synthetic peptide panel demonstrated similar remarkable stability without significant loss in reactivity or signal over the course of 60 days of testing, as illustrated in Supplementary Fig. 1.

To reduce the test complexity for POC use and increase its sensitivity for detecting both early and late LD patients, we evaluated whether combined IgM and IgG detection could be performed in the xVFA format. We screened a panel of secondary antibodies (Supplementary Table 1 and Supplementary Fig. 2) and found that by combining anti-IgM and IgG into a single assay, we could detect antibody binding to peptides at a higher rate than using IgM or IgG alone (Fig. 1g). By combining the detection of IgM, which appears in higher concentrations in the bloodstream earlier post-infection, along with IgG, the sensitivity of the LD test is improved. This is particularly important for early-stage disease diagnosis when current two-tier strategies have historically shown lower sensitivity.

Signals are read using a smartphone-based portable reader followed by automated processing of readouts to avoid bias in diagnostic interpretation (Fig. 1h)[22–24]. The custom reader consists of a tray that holds the paper assay device and slides into the reader, enabling the smartphone camera to capture images of the immunoreaction spots. Images are captured instantly and can be uploaded to the cloud for processing and automated analysis of signals. This feature can also promote interpretation of the results in a closed-loop setting, ensuring accurate reporting of tests to physicians and public health officials, and informing patient care in distributed health systems. The reader can be calibrated for use with any smartphone equipped with a camera

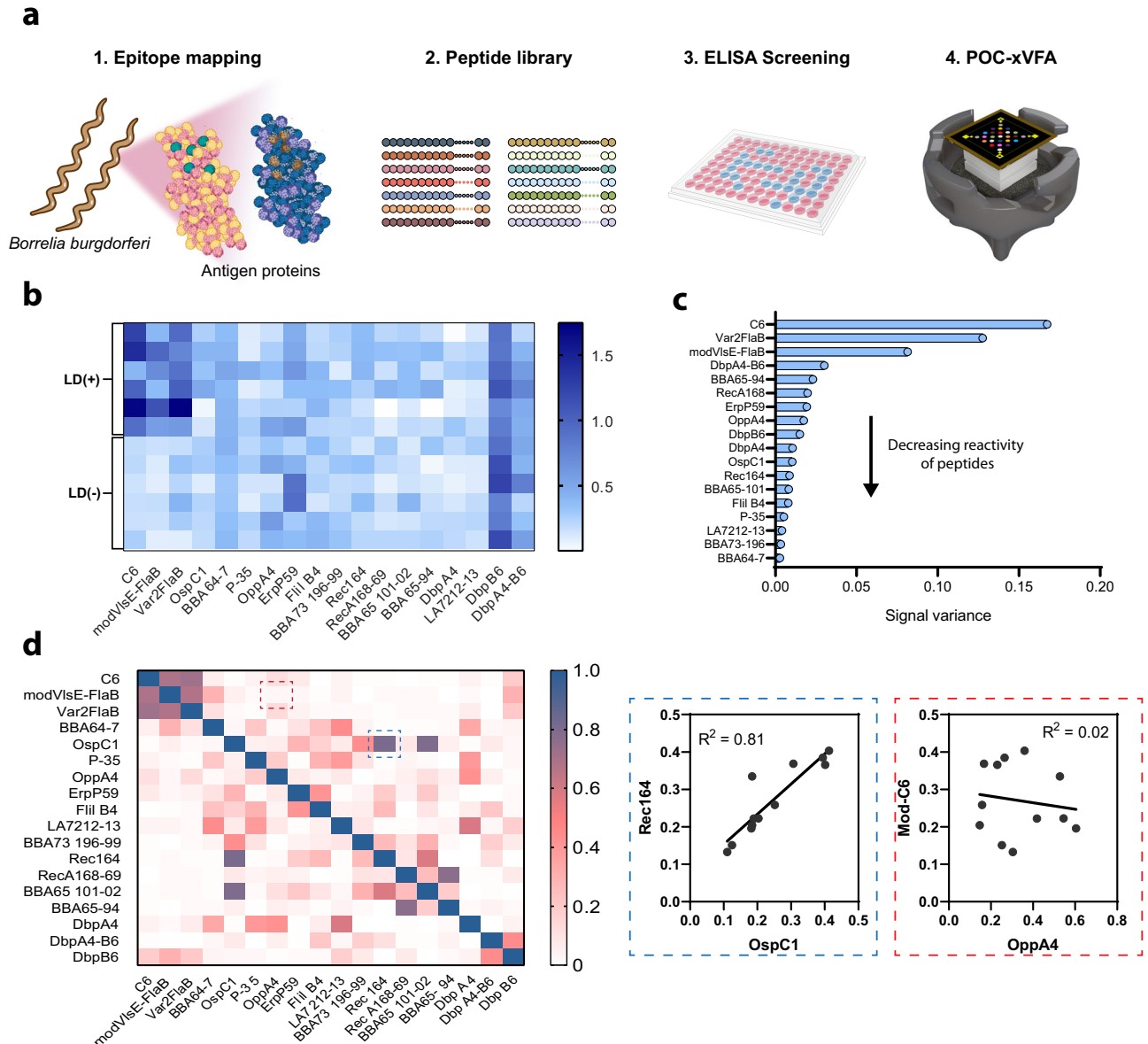

**Fig. 2 | *B. burgdorferi* antigenic peptide library screening and selection of prevalent LD-specific epitopes. a** Overview of the screening of epitopes to select the most relevant peptide antigens that function in the paper-based xVFA platform. **b** Heatmap representing the normalized signal intensities of peptides screened against patient samples positive (+) and negative (−) for LD, utilized in the development of the paper-based multiantigen xVFA platform. **c** Variance in the normalized signal intensity for each antigen peptide, determined through xVFA screening of control patient sera. The variance is plotted in descending order of reactivity, illustrating the comparative activity of the peptides against both positive and negative patient serum samples, highlighting the most active antigens. **d** Heatmap representing the correlation of peptides against each other. Inset shows examples of highly correlated peptides (Rec164 and OspC1) and non-correlated peptides (modVlsE-FlaB and OppA4), which yield additional information for a diagnostic panel. Panel **a** was created with BioRender.com released under a Creative Commons Attribution-NonCommercial-NoDerivs 4.0 International license (https://creativecommons.org/licenses/by-nc-nd/4.0/deed.en).

and is able to capture images with optimal illumination using green LEDs. The use of green LEDs for illumination within the reader, combined with a controlled distance between the sensing layer and the smartphone camera lens, enables the device to consistently capture high-quality images regardless of user training.

### Epitope mapping and multiplexed panel design
Protein antigens contain a combination of linear and conformational epitopes that can be unique to a given organism (specific epitopes) or commonly found in other antigens present throughout the biosphere (cross-reactive epitopes). To limit the cross-reactivity of antigen targets, we performed B-cell epitope mapping to identify epitopes specific to the *B. burgdorferi* proteins, OspC, DbpA, DbpB, BBK32, ErpP,

p35, OppA2, RecA, LA-7, FlilB, BBA64, BBA65, BBA66, and BBA73. Figure 2a shows an overview of our investigation approach to identify unique antigen epitopes of *B. burgdorferi* that could be used as diagnostic targets in a multiplexed assay. Synthetic peptides containing proposed linear epitopes were screened by ELISA, either as a part of this study (Supplementary Fig. 3) or in previous studies[18,25–29], using panels of sera from Lyme disease patients, healthy volunteers, and patients with other look-a-like diseases (See methods, Sample cohort 1). In this study, we also evaluated peptides containing two epitopes from different antigens, modVlsE-FlaB, DbpA4-B6, and Var2FlaB, which improved sensitivity in ELISA assays compared to individual peptides alone, while retaining the specificity by limiting non-specific binding[25,27]. Peptides that demonstrated sensitive and or specific

binding characteristics by ELISA were further screened using the xVFA to determine if differences in the nitrocellulose membrane substrate or unique flow-through format affected antibody affinity.

We measured the reactivity of the peptides immobilized on the nitrocellulose sensing membrane using a panel of well-characterized serum samples containing six LD-positive samples from different stages of the disease and six healthy control samples (See Methods, Sample cohort 2). Figure 2b presents a heatmap illustrating the normalized signal intensities for each peptide spot, as measured from the assay results. These intensities correspond to the signals detected and captured in images recorded by the smartphone-based reader. Peptides modVlsE-FlaB and Var2FlaB showed 100% sensitivity and specificity. DbpB6 and DbpA4-B6, which is a dipeptide combining epitopes from both DbpA4 and DbpB6, showed high reactivity even in healthy control samples, indicating non-specific interactions that were not observed in the ELISA format. Peptides ErpP59, OppA4, and BBA64-7 showed reactivity with four out of six positive serums tested, while also demonstrating non-specific interactions with one out of the six non-LD sera. A list of all peptides that were screened using the xVFA can be found in Supplementary Table 2. Figure 2c illustrates the variance in normalized signal intensities across both LD positive and negative samples on the sensing membrane, effectively representing the activity of the peptides. As previously identified, DbpA4-B6 and DbpB6 spots had a high variance in signal across patients, but this high variance was also connected to high background interactions against healthy samples and, therefore, was not selected for further test development. By plotting the correlation of the peptide signals across patients (Fig. 2d) we identified and removed correlated epitopes where antibodies against both epitopes were high or low in tandem and were already represented on the multiplexed panel. For example, both OspC1 and Rec164 were moderately reactive across LD patients and had a higher degree of correlation ($R^2 = 0.81$). Therefore, only OspC1 was included in the multiplexed panel as it is expressed early during the transmission of disease and is a marker for early-stage infection. Further, OppA4 was selected in the multiplexed panel as it was reactive with four out of the six LD patient sera and was uncorrelated ($R^2 \leq 0.4$ to all other peptides) with other peptides, indicating unique antibody recognition.

Following the initial analysis of antibody binding to 18 peptides, we downselected a subset of nine peptides based on specific criteria. We selected peptides that were reactive with at least two of the six LD control samples and exhibited higher variance in reactivity against the control samples, with the exception of DbpB6 and DbpA4-B6, which had general non-specific interactions across all samples. Additionally, peptides that showed lower inter-correlation, indicating the ability to capture unique antibody signatures, were included in the multiplexed panel. The nine selected peptides, with each spotted in duplicate along with three positive and four internal negative controls, constituted a total of 25 immunoreaction spots on our sensing array. This carefully curated multiplexed panel, which includes unique epitope-specific targets, enhances our capability to detect distinct epitope-specific immune signatures that single-plex tests may not detect. The selected peptides used for the training of our final xVFA diagnostic model were modVlsE-FlaB, Var2FlaB, BBA64-7, OspC1, OppA4, p35 21, ErpP59, FliI-B4, and BBA73 196-99.

**Training of xVFA peptide panel using early-stage LD samples**
To train the deep-learning diagnostic model, 40 out of the 70 serum samples from Sample cohort 3 were used (See Methods, Sample cohort 3). This included 20 early-stage LD samples and 20 healthy control samples collected by the Lyme Disease Biobank (LDB) from LD endemic regions (Supplementary Table 3). Samples were classified as either Lyme positive or negative using standard two-tier serology (STTT). All but one of the patient samples were positive for IgM western blot indicating early-stage infections. Meanwhile, only five

samples were positive and another five were equivocal for IgG western blot. Antibody positivity did not correlate with the presence or absence of EM, as EM-positive and EM-negative patients had a mix of IgM and IgG positivity. Further, the pool consisted of patients who reported experiencing a tick bite one to four weeks prior to the time of blood draw, thus ensuring that early onset LD disease samples were represented and used in the training of the developed assay. For training the diagnostic model, nine peptide targets derived from ten different antigens of *B. burgdorferi* were employed. Peptides were spotted in duplicate with positive and negative controls (Fig. 3a). Figure 3b illustrates the average normalized signal intensities calculated from two immunoreaction spots for each of the three xVFA replicate tests performed on the individual patient samples in the training subset (40 samples × 3 replicate, 120 xVFA tests). The reactivity of the antigen peptides varies across the LD and healthy control samples tested. Notably, the test showed reproducibility as the coefficient of variation (CV) of the average spot intensity per peptide across the three replicates was always below 10% (Fig. 3b).

We first investigated whether the signal from a single peptide would provide sufficient diagnostic performance for LD testing. The use of individual peptides in isolation fails to meet the performance requirements to serve as a single-tier test for Lyme disease diagnosis, with the best performance being 76% sensitivity and 95% specificity for the modVlsE-FlaB peptide (Supplementary Fig. 4). We then turned to a deep learning-based analysis of the multiplexed signals to form a synergistic model that combined information from the spots into a binary classification of either positive or negative detection.

A diagnostic model was trained using a deep-learning neural network to classify a sample as LD positive (prediction value >0.5) or negative (prediction value ≤0.5) using a cross-validation approach where 120 tests (40 patients × 3 replicates per patient) were used to optimize the architecture of the neural network model (see Methods). We used the model to first select the next most important immunoreaction spot to improve diagnostic accuracy as defined by the area under the curve (AUC) of the receiver operating characteristic (ROC). Figure 3c shows the peptide selection process using sequential forward feature selection (SFFS) and a comparison between the performance of different combinations of peptides as shown by the ROC curves. Here, each immunoreaction spot is considered a feature that serves as an input to the diagnostic model, totaling 25 different immunoreaction spots for nine peptide antigens plus control spots. The AUC for the different feature combinations is shown in Fig. 3d and yields a maximum when using a combination of three-peptide immunoreaction spots: modVlsE-FlaB, Var2FlaB, and OppA4. Using the combination of these three peptides, we achieved a sensitivity of 81.7% and a specificity of 96.7%, which exceeded the performance of the individual peptides or antigens in a standalone assay. modVlsE-FlaB, Var2FlaB, and OppA4 each yielded a sensitivity of 76, 76, and 7%, respectively, with specificity set at 95% for each. Figure 3e shows the prediction of our multiplexed diagnostic model using the three peptides selected with the SFFS method. Each dot represents the model output value for a separate xVFA test, where a threshold for positivity is set at 0.5. A clear separation is observed in model outputs for LD-positive and negative patients. The three samples (LDB08, LDB12, and LDB13) with a false negative xVFA test prediction (<0.5) were likely from patients with borderline antibody titers as evidenced by negative or equivocal results in the C6 Peptide ELISA, VlsE/pepC10 EIA, and IgG Western Blot. These three LD+ patient samples were from patients who did not present with an EM at the time of enrollment and remained negative or indeterminate by the IgG western blot. Additionally, two other positive samples (LDB17, LDB19) were negative for the C6 peptide ELISA and for the IgG western blot but were positive using xVFA. This suggests that having multiple epitopes represented can improve upon sensitivity when traditional single-target tests are not effective. One of the three replicates from two different healthy control samples

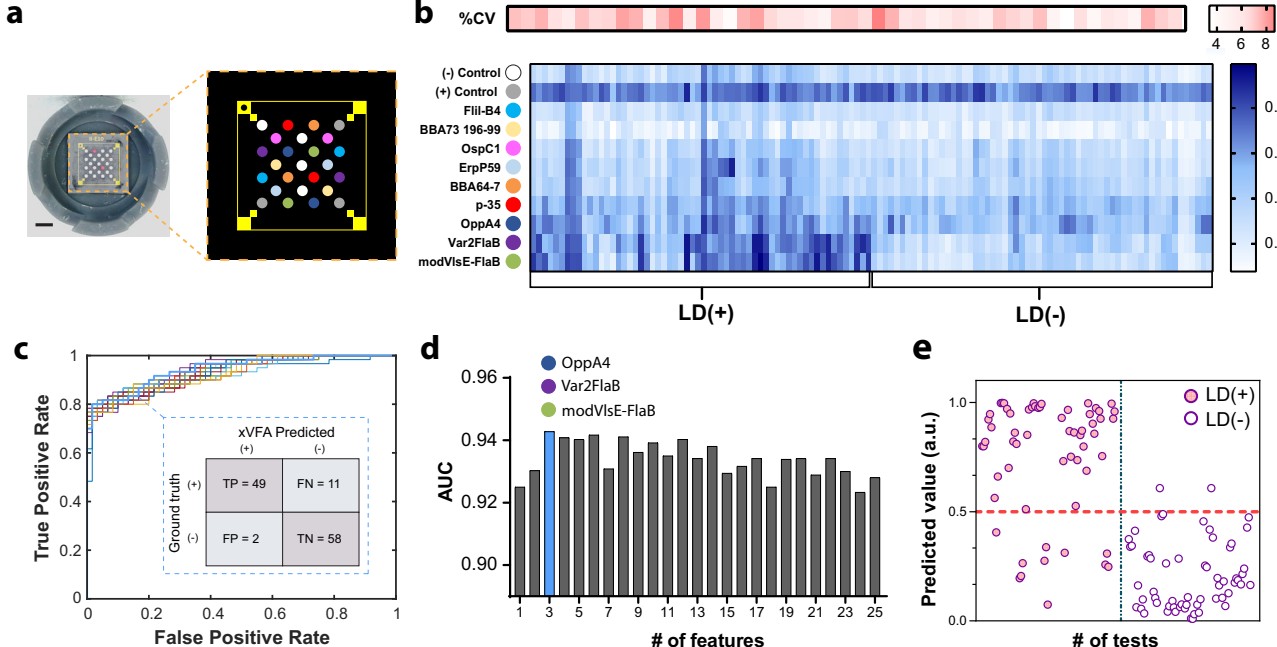

**Fig. 3 | Training of the multiantigen xVFA panel using early-stage LD patient samples obtained from the Lyme Disease Biobank (LDB). a** The multiantigen coated sensing membrane and map of each antigen location spotted in duplicates along with the positive and negative control reaction spots. **b** Heatmap displaying the normalized signal intensities calculated from two immunoreaction spots for each of the three xVFA replicate tests performed on the individual patient samples in the training subset (40 samples × 3 replicate, 120 xVFA tests). The color scale on the top represents the average %CV in measurement for all peptides per patient sample. **c** Receiver operator characteristics (ROC) resulting from the neural network-based multiplexed diagnostic model comparing the model's performance when different numbers of peptides are used to train the network. The inset shows the confusion matrix for a three-peptide model. **d** Bar plot showing the area under the curve (AUC) for the different input features used to train the diagnostic model. A three-peptide model including single immunoreaction spots from modVlsE-FlaB, Var2FlaB, and OppA4 yields the highest AUC of 0.94. **e** The final prediction outcome from the neural network-based diagnostic model, indicated as values from 0 to 1, where each dot represents a test that was performed on the xVFA using a patient sample (40 patient samples tested in triplicate). The red dotted line at 0.5 indicates the threshold for positivity of the diagnostic model.

was misclassified as positive, with a prediction value that was closer to the 0.5 threshold. The tests had higher background interaction with all the peptides resulting in a final prediction value that was higher than the threshold of 0.5. The xVFA was able to correctly identify the samples from the training cohort as either LD positive or negative with similar accuracy (~90%) compared to the first-tier centralized lab-based C6 peptide ELISA (Oxford Immunotec, Marlborough, MA). Additionally, the xVFA outperformed the second-tier IgM and IgG western blot (Viramed; Biotech AG, Germany), which had an accuracy of 84 and 66%, respectively.

## Blinded assessment of single-tier xVFA diagnostic model using diverse CDC cohort

Using the optimal diagnostic model outlined earlier, we assessed the specificity of our single-tier multiplexed test using a diverse set of samples obtained from the CDC Lyme disease repository, including patients with different stages of LD, healthy control groups from both endemic and non-endemic regions, and patients with look-alike disease with no previous exposure to LD. The LD samples included early acute stage, early convalescent stage, and late-stage disease cases. Early acute-stage disease cases tested negative using STTT, while early convalescent and late-stage cases were confirmed positive by STTT. Figure 4a shows the prediction values of the trained machine-learning diagnostic model for the CDC samples tested in triplicate. Table 1 summarizes the performance of the xVFA along with the results from the individual two-tier reference tests and two-tier diagnosis on the same samples enabling a direct comparison. The CDC panel samples were tested in triplicate using the xVFA. The xVFA successfully detected all the early convalescent and late-stage LD samples. In repeat testing of the xVFA, one replicate failed to detect either a convalescent

or late-stage LD sample, which could be due to variations in testing. During the development of the peptide-based panel, we hypothesized that developing a peptide-based assay and limiting cross-reactive epitopes would enhance the specificity. Notably, the single-tier xVFA showed no cross-reactivity with look-alike diseases or healthy control samples, while the reference tests that constitute the individual single tiers of the two-tier assay exhibited non-specific reactivity with both these groups (Table 1). Both the single-tier xVFA and the STTT did not detect any of the acute LD samples in the panel. This could be attributed to the low LD-specific antibody levels in this particular cohort of early acute LD samples, requiring follow-up testing until a detectable immune response can be measured using these tests. Only the modified two-tier testing algorithm performed slightly better using the IgM-based diagnosis and was able to classify one additional acute LD sample as LD positive.

As represented in Fig. 4b, the xVFA demonstrated 100% agreement (Cohen's Kappa coefficient (κ) of 1.0) with the STTT algorithm and 96.6% agreement (κ of 0.92) with the modified two-tier testing (MTTT) algorithm when evaluated using the CDC panel, a sample set previously unseen by the diagnostic model, indicating near-perfect agreement. When comparing the performance of the single-tier xVFA with the individual centralized lab assays, we found that the xVFA performs better than many of the individual laboratory assays. Figure 4c compares the accuracy of the xVFA along with the individual reference tests and the standard and modified two-tier algorithms. The xVFA demonstrated an accuracy of 88% (95% CI: 76.7–99.2%), correctly identifying 8 of the 12 LD cases as positive and all the 20 non-LD cases as negative. On the other hand, the individual reference tests either failed to detect some of the LD cases or falsely identified some of the non-LD control samples as positive (Table 1), indicating a lack of

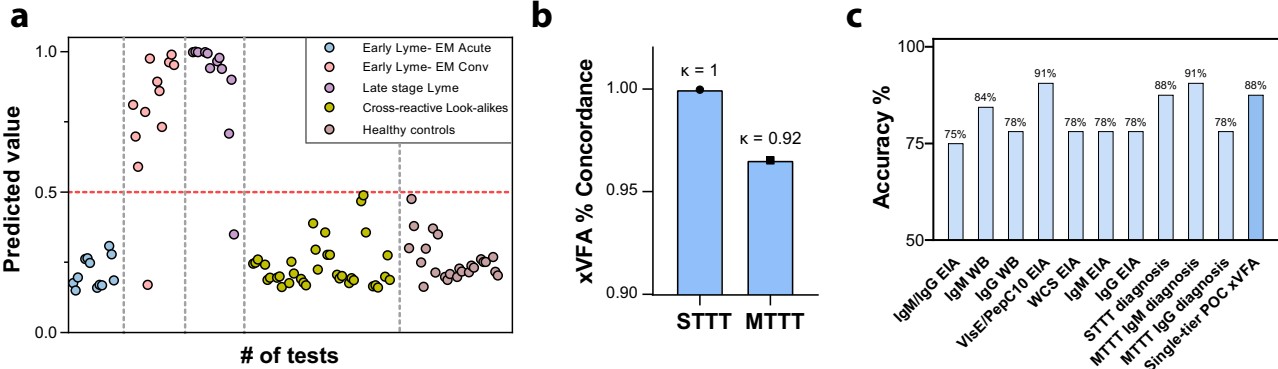

**Fig. 4 | Assessment of diverse samples from the CDC using the multiantigen xVFA and the optimized neural network-based diagnostic model. a** Prediction outcome from the neural network-based diagnostic model using the trained network indicated as values from 0 to 1, where each dot represents a test that was performed on the xVFA using patient samples that had different stages of LD, healthy control samples from regions where LD is endemic and non-endemic, cross-reactive samples from look-alike diseases such as fibromyalgia, rheumatoid arthritis, multiple sclerosis, mononucleosis, syphilis, and severe periodontitis. The red dotted line at 0.5 indicates the threshold for positivity of the diagnostic model. Three xVFA tests were run per sample. **b** Bar plot displaying the percentage concordance of the xVFA assay with the standard two-tier testing (STTT) and modified two-tier testing (MTTT) IgM diagnosis. **c** Bar plot comparing the accuracy of the single-tier xVFA assay with individual centralized lab-based STTT and MTTT reference tests, including the overall diagnostic outcomes using each algorithm.

**Table 1 | Clinical samples obtained from the CDC to assess the specificity of the xVFA and the corresponding ground truth information**

| Sample group | Standard two-tier tests (STTT) | | | Modified two-tier tests (MTTT) | | | | Diagnosis | | | |
|---|---|---|---|---|---|---|---|---|---|---|---|
| | IgM/ IgG EIA | IgM WB | IgG WB | Zeus VlsE/ pepC10 EIA | Zeus WCS EIA | Zeus IgM EIA | Zeus IgG EIA | STTT | MTTT IgM | MTTT IgG | xVFA |
| Early Lyme-acute | 0/4 | 1/4 | 0/4 | 2/4 | 1/4 | 1/4 | 0/4 | 0/4 | 1/4 | 0/4 | 0/4 |
| Early Lyme-convalescent | 4/4 | 4/4 | 1/4 | 4/4 | 4/4 | 4/4 | 1/4 | 4/4 | 4/4 | 1/4 | 4/4 |
| Late Lyme disease | 4/4 | 2/4 | 4/4 | 4/4 | 4/4 | 4/4 | 4/4 | 4/4 | 4/4 | 4/4 | 4/4 |
| Look-alike diseases | 2/12 | 0/12 | 0/12 | 1/12 | 1/12 | 1/12 | 0/12 | 0/12 | 0/12 | 0/12 | 0/12 |
| Healthy control | 2/8 | 0/8 | 0/8 | 0/8 | 3/8 | 3/8 | 0/8 | 0/8 | 0/8 | 0/8 | 0/8 |

sufficient specificity within these individual two-tier tests. For example, the reference tests, STTT IgM/IgG EIA, Zeus VlsE/pepC10 EIA, Zeus WCS EIA, and Zeus IgM EIA all showed cross-reactivity with either Syphilis, an infection caused by a spirochete known to cross-react with LD tests, or with other healthy control samples. It is only when the individual two-tier tests are combined that the specificity is improved, highlighting the potential for erroneous reporting of the individual assays, and the current necessity for the two-tier algorithms. A single-tier POC LD test with high accuracy can provide a rapid and reliable approach to detect LD, improving patient outcomes.

**Validation of the xVFA assay using early-stage LD samples**

We validated the xVFA assay and the diagnostic performance with a second cohort of 30 samples from the LDB (See methods, Sample cohort 3). This included 15 early-stage LD samples and 15 healthy control samples collected by the LDB from LD endemic regions (Supplementary Table 4). All the patient samples in this subset were positive for IgM western blot, indicating early-stage infections, while only three were positive, and another four were equivocal for IgG western blot. Figure 5a displays the prediction values for these samples obtained using the optimized diagnostic algorithm, with each sample tested three times to show consistency across replicates. A clear distinction can be observed between the LD positive and LD negative samples, with the majority of LD positive samples exhibiting prediction values above the threshold, denoted by the red dashed line. The xVFA correctly identified all the early-stage infection samples as LD positive, indicating an effective performance from the assay even

during early immune response. A total of 90 tests (30 patients × 3 replicates per patient) were used to validate the optimized model. The diagnostic accuracy of the xVFA is summarized in Fig. 5b, which presents a confusion matrix contrasting the xVFA predicted results with the ground truth for the tested samples. The model showed remarkable diagnostic precision, yielding a high sensitivity of 95.5% (95% CI: [89.5%, 100%]) and a specificity of 100%. Notably, the assay did not yield any false positives, and only one of the three replicates from each of the two LD cases were incorrectly categorized as false negatives, demonstrating the assay's high reliability in distinguishing early-stage LD infections. This performance underscores the xVFA's potential to be used as a single-tier LD assay, offering rapid test outcomes with reliable precision, which is crucial for accurate and timely diagnosis.

## Discussion

Synthetic peptides are excellent capture antigens as they preserve the binding sites for the detection of anti-*B. burgdorferi* antibodies while reducing the cost and complexity of diagnostic tests. Peptides can be chemically synthesized at extremely low cost and on a scale to manufacture tests in bulk. Native, or recombinant proteins of *B. burgdorferi* are harder to synthesize, purify and isolate, leading to higher costs, particularly in downstream quality control (Supplementary Table 5), and have a reduced shelf-life before test performance is impacted (Fig. 1f). Recombinant proteins may also contain cross-reactive epitopes, which limit the performance, reducing specificity, as seen in our previous study using a multiplexed panel of *B. burgdorferi* antigens and a single peptide[19,30]. Additionally, by selecting epitopes that are highly

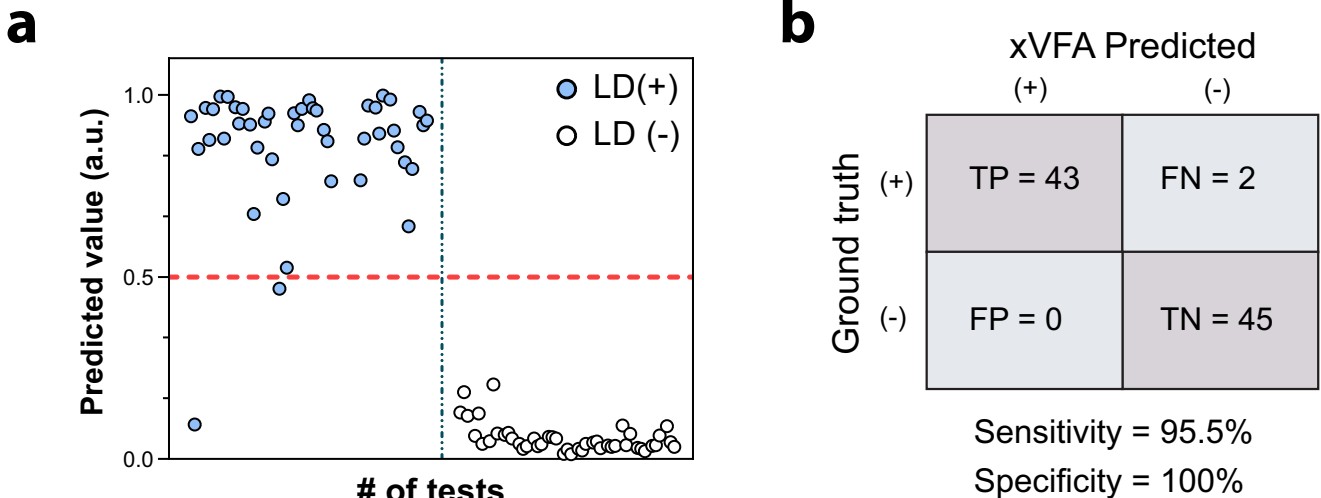

**Fig. 5 | Validation of the single-tier xVFA and the optimized diagnostic model using samples from the Lyme Disease Biobank in cohort 3. a** Prediction outcome from the neural network-based diagnostic model using the trained network indicated as values from 0 to 1, where each dot represents a test that was performed using the xVFA (30 samples × 3 replicates, 90 xVFA tests). The dotted horizontal line represents the threshold for a positive diagnosis. The cohort consisted of 15 LD positive samples and 15 LD negative samples that were confirmed using standard two-tier testing for ground truth. **b** Confusion matrix summarizing the performance of the xVFA in terms of true positives, false negatives, false positives, and true negatives, with calculated sensitivity and specificity. The assay demonstrates high diagnostic precision, validating its effectiveness for Lyme disease detection.

conserved among different strains of *B. burgdorferi* with limited homology to sequences in other antigens likely to be encountered by humans, we can reduce cross-reactivity. Through the inclusion of multiple epitopes with our machine learning framework, we can maintain or improve sensitivity compared to current assays, while having a limited epitope pool compared to a whole recombinant protein or bacterial sonicates. Starting with a set of 18 curated peptides, we further downselected using our initial screening data set to eliminate peptides that did not bind well to paper, were highly correlated to other peptides, or were generally reactive to LD-negative sera. With a smaller set of nine peptides, we screened a cohort of 40 samples to develop a training data set for further refinement of a diagnostic model to maximize sensitivity while avoiding false-positive results. Our final model selected three immunoreaction spots (three different peptides, modVlsE-FlaB, Var2FlaB, and OppA4, representing four different antigens).

Although current lab-based EIAs are used as part of a two-tier assay, if EIAs are used as single-tier tests, there was a tradeoff of sensitivity vs. specificity in comparison to the xVFA, as evidenced by their performance in the CDC cohort. For example, the Zeus IgM EIA, WCS EIA, and other Zeus VlsE/pepC10 EIA all detected one or two of the acute-stage samples but were also found to non-specifically bind to look-alike disease samples and healthy patient samples from endemic and non-endemic areas, reducing specificity and the positive predictive value of these centralized lab tests. Similarly, the Zeus IgG EIA did not cross-react with any of the look-alike disease or healthy control samples, but only detected 5 out of the 12 LD cases in the CDC panel. Both the standard and modified two-tier serology-based diagnostic algorithms comprise individual tests that have varying performances that may lead to compounded inaccuracies in diagnosing LD. For instance, within the standard two-tier framework, the VIDAS Lyme IgM/IgG, IgM Western blot, and IgG Western blot tests each detected only 8, 5, and 5 of the 12 LD cases in the CDC panel, respectively (Table 1). The low sensitivity of the tests, combined with the tiered approach of testing, can potentially lead to a decline in overall sensitivity, increasing the likelihood of missing true positive cases when tested across a larger patient cohort. Furthermore, the first-tier VIDAS Lyme IgM/IgG assay incorrectly identified 4 out of the 20 non-LD cases in the CDC cohort, increasing the probability of false-positive results.

This necessitates further confirmatory tests under the two-tier paradigm, thereby increasing the complexity, cost, and overall burden of diagnosing LD. Similarly, the modified two-tier framework comprises tests with similar limitations, with the first-tier test detecting 10 out of the 12 LD cases while the second-tier IgM and IgG EIA detecting only nine and five LD cases, respectively. Additionally, the VlsE/pepC10 EIA, a first-tier assay, incorrectly identified one non-LD sample as positive, while the Zeus IgM EIA misclassified four out of the 20 non-LD samples in the CDC panel as LD positive. Unlike the two-tier approach, where the low specificity of the individual reference tests necessitates additional follow-up testing and management of false-positive patients, a single-tier assay could reduce the complexity of diagnosis, offering rapid disease confirmation, potentially leading to cost savings and timely treatment.

Finally, the xVFA achieved a sensitivity of 95.5% and a specificity of 100% when evaluated using a previously untested subset of 30 samples from the LDB cohort, utilizing the optimized diagnostic model. This particular subset included samples from early-stage infections evidenced by IgM dominant immune responses, and lack of seroconversion to IgG. Additionally, the synthetic peptide-based xVFA did not indicate any non-specific reactivity with healthy control samples from this cohort, even though they were acquired from regions where LD is endemic. The high specificity of our xVFA combined with the comparable sensitivity in a POC-compatible format would be suitable in a rapid test setting such as local pharmacies, clinics, or other POC settings, as per CLIA guidelines and regulations, enabling healthcare providers to quickly determine appropriate treatment with antimicrobials upon obtaining a positive result. A negative result may indicate repeat testing is needed in acute patients with other signs and symptoms after waiting for serum IgM and/or IgG antibody levels to increase further. It is possible that patients with previous LD infections who had recovered may have remaining low levels of IgG that would yield a false-positive test. Future guidelines could inform patients or healthcare providers to interpret the test results in the context of the patient's history and other signs and symptoms. The xVFA test is optimized for use with serum samples, which are obtained by removing blood cells from patient blood. The isolation of coagulants from the blood aids in analysis and results in a higher antibody concentration sample that is conducive to the detection of early disease.

**Table 2 | Comparison of the two-tier and single-tier xVFA assay with gold standard serology tests**

| | | | Conventional laboratory tests | | Previous assay [19] | Current assay |
|---|---|---|---|---|---|---|
| | | | Two-tier test (STTT) | Two-tier test (MTTT) | Two-tier xVFA | Single-tier xVFA |
| Assay specifications | Test format | | Two-tier | Two-tier | Two-tier | Single-tier |
| | Antigen type used | | Proteins | Proteins/peptide | Proteins/peptide | Peptide |
| | IgM or IgG detection | | IgM and IgG individually | IgM and IgG individually | IgM and IgG individually | Combined IgM/IgG |
| | Point of care? | | No | No | Yes | Yes |
| | Cost | | >$400¶ | >$400¶ | $25.98 | $2.59 |
| Performance | Bay Area Lyme Disease Biobank | Sensitivity | NA | NA | 90.5% | 95.5% |
| | | Specificity | NA | NA | 87.0% | 100.0% |
| | CDC Lyme Disease Biobank | Accuracy | 88% | 91% | NA | 88.0% |

[19] ACS Nano 2020, 14, 1, 229–240.
Cost of commercial two-tier tests.

When deployed for POC testing or testing in resource-limited regions, the xVFA could be coupled with a rapid serum extraction step to generate 20 μL of serum or plasma that can be utilized for testing on-site. Table 2 summarizes the advantages of the proposed single-tier xVFA for Lyme disease detection, compared to two-tier conventional laboratory assays and the two-tier xVFAs using proteins and peptides as a multiplexed panel.

The POC-compatible xVFA, utilizing the three-peptide antigens identified by our model, is expected to be considerably more cost-effective than traditional EIAs, while also delivering superior enhanced performance. The single-tier assay also demonstrated a CV of 8.5% between different tests by the same operator and a CV of 9.3% for tests performed by different operators who are not specialized in clinical testing (undergraduate students), ensuring repeatable results independent of operator training. The material cost associated with a single xVFA is less than $3 and can be manufactured at scale owing to compatibility with automated spotting equipment commonly used in LFA production (Supplementary Table 5). The utilization of a selected subset of three peptides in the xVFA implies that the majority of immunoreaction spots on the platform are not utilized. Future work could investigate whether replacing these spots with copies of the selected peptides would lead to even further improved performance. Alternatively, epitopes for other tick-borne illnesses or other related bacterial pathogens could be included to expand the diagnostic capabilities or monitor other epitope-specific antibodies from patients and acquire additional data on the prevalence of different immuno-phenotypes across a significantly expanded set of patients. Using diagnostic tests to derive epitope-specific community health information would serve to understand the underlying disease yielding critical information on LD pathogenesis and better formulation of treatment measures. Large data sets that leverage multi-epitope arrays associated with clinical outcomes could also be used to train even better models to diagnose LD or develop region-specific models, if required.

Cloud connectivity with a smartphone implementation can enable the integration of Lyme diagnostic results with patient care and public health guidance. The monitoring of new positive disease incidences could also enable tracking of tick-borne diseases informing public health guidance in areas of high endemicity. Smartphone-based interpretation also eliminates bias in results interpretation by serving as a seamless interface between the diagnostic test and the neural network-based prediction model. The reported test, combined with recent advances in telemedicine and at-home diagnostics, coupled with smartphone-based interfaces, could enable more efficient test-to-treat paradigms, seamlessly integrating physician input for prescription, therapeutic delivery, and ultimately leading to more rapid and effective patient care.

## Methods
### Ethical statement
The study included deidentified and blinded patient serum samples that were acquired by Lyme Disease Biobank with informed consent under Institutional Review Board (IRB) approval Advarra IRB protocol Pro00012408. All experiments are in compliance with the University of California, Los Angeles's institutional review board.

### Overview of the multiplexed vertical flow device (xVFA) platform
The vertical flow assay platform consists of paper layers that are stacked together to form a three-dimensional fluidic network that transports assay fluids vertically by capillary wicking action. The paper layers are housed inside a plastic cassette that can be separated into two parts through a twist mechanism, revealing a multiplexed sensing membrane on the top of the bottom section, containing 25 immuno-reaction spots spatially isolated by a hydrophobic wax-printed barrier. Each spot is preloaded with different capture peptides (1 mg/mL) and a Goat anti-Mouse IgG (0.1 mg/mL) (#1036-01, SouthernBiotech, AL) that acts as the positive control spot for the assay to ensure that the gold nanoparticle conjugated mouse anti-human secondary antibodies successfully flowed through the xVFA and bound to the sensing membrane. The top section of the device contains paper layers that are engineered to ensure a uniform flow of samples and reagents across the entire cross-section of the sensing membrane. The bottom consists of highly efficient absorption pads below the sensing membrane that act as a sink for the assay fluid. For each assay, we use two top sections of the device during operation. The first top introduces the sample into the sensing membrane, followed by incubation, which ensures the binding of *B. burgdorferi*-specific antibodies to the different peptide antigens immobilized. The second top is used for signal generation by the addition of anti-human IgM (#9022-01, SouthernBiotech, AL) and IgG (#9040-01, SouthernBiotech, AL) antibodies conjugated to gold nanoparticles (AuNPs) (#15707, Ted Pella Inc., CA). The AuNPs bind to the LD-specific IgM and IgG antibodies that were captured and the non-specific signals are washed away by buffer solution. The xVFA operation was optimized by tuning the running buffer components, sample volume, and pore size of the nitrocellulose membrane using control LD positive and healthy samples (Supplementary Fig. 4). Upon completion of the assay, the sensing membrane is imaged using a low-cost portable smartphone reader that captures the individual reaction intensities under a green illumination for maximum light absorbance and signal-to-noise. The normalized signal intensity is calculated by dividing the post-assay signal by the pre-assay background, and then subtracting this fraction from 1. This normalizes the raw intensities of the reaction spots against their pre-assay background. Subsequently, the resulting normalized signal

intensity is analyzed through a machine learning-based multiplexed diagnostic algorithm to determine final prediction values and seropositivity of the tested sample. The entire assay and the diagnostic inference based on the different immunoreaction spots can be completed in under 20 min. The overall material cost for the entire consumables is less than $0.4 per test (Supplementary Table 5).

## Assay operation

To perform the xVFA assay, the first step is to record a background image of the unused sensor using the portable smartphone reader. The device is then assembled by attaching the first top case to the bottom case with a simple twist and activated by adding buffer solution and 20 μL of serum sample. After the buffer and serum are fully absorbed into the xVFA cassette (8 min), the first top case is opened and replaced with the second top case for signal generation. The 40 nm AuNPs-conjugated detector antibody solution (a mixture of mouse anti-human IgM and IgG in a 1:1 ratio) and running buffer are introduced into the device and incubated for an additional 8 min. During this time, the AuNP conjugated to the detector antibodies will react with human anti-borrelia immunoglobulins, producing an intense signal. Once the incubation period is complete, the xVFA cassette is opened and an image of the multiplexed signal on the paper membrane is captured using the mobile-phone reader.

## Smartphone reader

The portable assay reader consists of a smartphone (LG G4H810) with a 3D-printed opto-mechanical attachment that contains four 525 nm wavelength light-emitting diodes (LEDs) for uniform illumination of the sensing membrane. An external lens is mounted below the built-in camera lens of the smartphone within the 3D-printed attachment. All images were taken in raw dng format using the standard Android camera app on the smartphone. To measure the signals, the bottom case is connected to a 3D-printed tray and slid into the reader for easy and repeatable measurements. The cost of the reader components combined with the smartphone is less than $200.

## Image processing and neural network-based analysis

To analyze the results of the assay, raw dng images of the sensing membrane taken before (background image) and after (signal image) the assay are first converted to tiff format. The green pixels are then extracted, and the background and signal images are registered to each other using a rigid transformation. The immunoreaction spots are identified in the background image, and a fixed-radius mask is defined for each spot, covering ~80% of the immunoreaction spot area. The pixel intensity within this mask is then calculated for the registered signal image and normalized by the average pixel intensity of the corresponding immunoreaction spot in the registered background image. This background normalization procedure helps to compensate for nonuniformities in illumination and local defects that might exist within the immunoreaction spots on each xVFA sensing membrane. The immunoreaction spots functionalized with nine different capture antigens in duplicate are used as input for the deep-learning analysis. This helps to accurately analyze and interpret the results of the combined IgM and IgG antibody measurement for individual immunoreaction spots. The image processing algorithm can read and interpret two images per second, providing almost instantaneous results and ensuring a rapid outcome.

The decision neural network contains an input layer with M nodes (e.g., M = 25 with each immunoreaction spot on the multiplexed panel including positive and negative control spots), three fully connected hidden layers with 128, 64, and 32 nodes, in the first, second, and third layers, respectively. Each layer contains batch normalization, a 50% dropout, and a rectified linear unit (ReLU) activation function, except for the final output layer, which uses a sigmoid activation function, yielding a network output as a numerical value between 0 and 1. A final

binary classification is then made by evaluating the numerical output with the blind cut-off value of 0.5. The training was carried out using a binary cross-entropy loss function with Adam optimizer along with a learning rate of 0.0001 and batch size of 20. The architecture was determined by carrying out a grid search optimization of the major neural network hyperparameters, including the number of layers, number of units per layer, regularization, dropout, and learning rate.

Optimized neural network architecture was used during the selection of the optimal subset of immunoreaction spots. This spot selection process was implemented using a training set of serum samples (i.e., 40 serum samples from LDB) through the SFFS process, where the signals from each sensing channel were added one at a time into the input layer of the neural network and then trained via k-fold (k = 5) cross-validation. After the addition of each input feature, the performance of the network was evaluated based on the area under the curve (AUC) scores and the input feature that yielded the best network performance for that iteration was then kept as an input feature until all 25 immunoreaction spots were included as an input (Fig. 3d). The optimized diagnostic model, which incorporated three peptides including modVlsE-FlaB, Var2FlaB and OppA4, was trained using only the 40 serum samples from LDB. This model was used for validation and assessment of the single-tier xVFA across all other samples tested in this study without any modification, including the 30 serum samples in the separate validation cohort from LDB and the 32 serum samples tested in the CDC cohort.

## Stability of antigens and shelf-life testing

To assess the stability of the peptide and protein antigen targets after immobilization on the nitrocellulose membrane, sensing membranes were spotted with 1 mg/mL of modVlsE-FlaB peptide and VlsE native B. burgdorferi protein and immobilized in three immunoreaction spots each. These membranes were then tested using control Lyme disease-positive and healthy samples at various time points between days zero and ninety. The signal intensities corresponding to the peptides and proteins were compared to determine the stability of the respective antigens. Further, the shelf-life of the combined synthetic peptide panel, spotted on the nitrocellulose membrane, were tested over time to determine the consistency of performance of the full assay over 60 days of testing at various time points. Each immunoreaction spot was coated with either 1 mg/mL of the respective peptide, goat anti-mouse IgG as positive control or PBS buffer with 1% BSA as negative control, as shown in Fig. 3a. All xVFA cases for stability and shelf-life measurements were prepared and stored at 25 °C until tested. The results from each individual immunoreaction spot were processed through the optimized machine-learning diagnostic model, which yielded the prediction values for each xVFA test. These values were then used to assess the assay's consistency and reliability over the evaluation period.

## Epitope mapping

Linear B-cell epitope mapping was performed by ProImmune, Inc., using their ProArray Ultra Custom Microarray. Overlapping peptide libraries were generated for the indicated proteins consisting of 15-mer synthetic peptides overlapping by ten amino acids (AA) (5AA-offset). The peptides were printed as arrays on glass slides, and the arrays were probed with five dilutions of serum from eight patients considered highly seropositive for Lyme disease (9–10 bands positive on an IgG western blot). Positive binding was identified with fluorescently labeled anti-human IgM, IgG, IgA antibody (or just IgG for the paralogous proteins, BBA64, 65, 66, 73). An epitope was considered for further evaluation if a minimum of 6 of 8 sera (75%) demonstrated positive binding in two or more serum dilutions. Epitopes with high (>75%) sequence conservation among different strains of B. burgdorferi and low (<50%) sequence homology to unrelated antigens were prioritized. Prospective epitopes were synthesized as individual

peptides of up to 45 AA in length (some epitopes overlapped multiple spots on the array). Epitope-containing peptides were further screened by ELISA using larger serum sets containing serum from patients with EM lesions, Lyme arthritis, syphilis, rheumatoid arthritis, and serum from healthy volunteers living in Lyme disease endemic or non-endemic areas.

## Peptide selection using ELISA

Peptides were coated on 96-well MAXISorp plates at a concentration of 10 μg/ml in 0.1 M sodium carbonate, pH9.4 for 1 h at room temperature. After 1 h bout 250 μl of 1% BSA (Sigma) in PBS (blocking buffer) were added to each well, and plates were stored at 4 °C overnight. Plates were washed with 0.05% Tween 20 in PBS using an Aquamax automated plate washer (Molecular Diagnostics). 1:100 dilutions of patient serum in blocking buffer (50 μl/well) were added for 2 h at room temperature. Plates were washed as above, and a 1:15,000 dilution of goat anti-human IgG, IgA, and IgM (Jackson Immunoresearch) was added (50 μl/well) for 1 h at room temp. Plates were washed and 50 μl of TMB (KPL) was added per well for 30 min in the dark. Reactions were stopped with the addition of 25 μl of 2 N $H_2SO_4$. The color change was read on a Molecular device Spectramax. Data were presented in Supplementary Fig. 3 as absorbance at 450 nm.

## Peptide selection using xVFA

To screen the peptides selected using ELISA, we spotted 0.8 μL of 1 mg/mL of each peptide on a 25-spot sensing membrane in duplicates. We tested each peptide with all the samples from cohort 1 consisting of LD positive and healthy control samples. The signal intensities obtained were compared and the peptides were ranked based on their sensitivity, specificity, and correlation in identifying similar signatures of the patient samples.

## Clinical samples

Serum samples were obtained with unidentified labels with the sample information blinded until completion of the xVFA assay unless otherwise mentioned. The xVFA results were shared with the biobanks following which the labels were unblinded and identifiers were used to validate the performance.

Sample cohort 1 was used for epitope mapping and identification of *Borrelia burgdorferi*-specific targets and consisted of Lyme disease sera used for peptide screening by ELISA and were banked samples accumulated by Biopeptides, Corp. over the course of the last 30 years. They were collected from patients under informed consent with the approval of the institutional review boards of Stony Brook University in Stony Brook, NY, New York Medical College in Westchester, NY, and Gundersen-Lutheran Medical Center in La Crosse, WI. The 50 samples from Gundersen-Lutheran Medical Center in La Crosse, WI, had a clinician-documented EM lesion of >4 cm, appropriate epidemiologic history (e.g., tick bite or exposure), and were seropositive according to a whole-cell ELISA (VIDAS by BioMerieux, Durham, NC, USA). We were not provided with the clinical laboratory results for the other deidentified Lyme disease patients beyond the fact that the patients were EM+. A total of 20 late Lyme disease samples were collected from patients at Gundersen-Lutheran Medical Center in La Crosse, WI, with Lyme arthritis (LA) (n = 20) that had one or more episodes of swollen joints, appropriate epidemiologic history, and positive reactivity using a whole-cell ELISA (VIDAS). The regions where the samples were collected, lower New York and Wisconsin, are highly endemic areas for Lyme disease. Sera from healthy volunteers were collected in New Mexico, which is not endemic for Lyme disease, and were purchased from Creative Testing Solutions (Tempe, AZ, USA). Sera from patients with rheumatoid arthritis (RA, rheumatoid factor status unknown) or who were Rapid Plasma Reagin positive (RPR+, first-tier test for syphilis) were purchased from Bioreclamation LLC (Westbury, NY, USA). These samples were collected in a region endemic to Lyme

disease (the northeastern US). A total of 34 of the 35 RPR+ sera used in this study had positive or equivocal antibody levels against *Treponema pallidum* by ELISA (Abnova, Walnut, CA, USA). Some serum samples are not represented in all data sets because they were fully consumed during experimentation.

Sample cohort 2 was used for screening of the peptides on the xVFA assay platform and included 12 clinical samples consisting of six Lyme disease-positive samples from patients with different stages of LD purchased through a commercial vendor LGC diagnostics and six healthy control samples that were collected by the LDB from regions where LD is endemic.

Sample cohort 3 consisted of 70 samples from the LDB and was used for both training and validation of the deep-learning diagnostic model (Supplementary Table 3). The cohort was split into two subsets for training and validation of the xVFA and the optimized deep-learning algorithm. The training subset included 20 two-tier positive LD samples and 20 healthy control samples that were negative using two-tier serology. The validation subset consisted of 15 LD positive and 15 LD negative samples that were also confirmed with two-tier serology. Of the 70 patient samples in this cohort, 40 training and 10 validation samples were analyzed on the same day. The remaining 20 samples, which were a subset of the validation cohort, were tested 18 months after the initial training of the optimized diagnostic algorithm. The LDB samples were collected from East Hampton, NY, between 2014 and 2019 (EH), and Marshfield, WI, from 2016 to 2019 (WI)[31]. Samples were collected through LDB sponsor protocol with Institutional review board (IRB) approval Advarra IRB protocol Pro00012408. Within the LDB sample cohort, there was a nearly even gender distribution, with 48.3% females and 51.3% males. The cohort included 35 two-tier positive Lyme disease samples with signs and symptoms of early Lyme disease. Patient samples in this group were sourced from individuals in Lyme disease-endemic regions, with EM lesions diagnosed by a physician when present, irrespective of size. Participants also included those from endemic areas presenting with symptoms like headache, fatigue, fever, chills, or musculoskeletal pain, or with a history of suspected tick exposure. Additionally, 35 healthy control samples were collected from regions where Lyme disease is endemic, with at least two years of recorded residency in the region. The overarching exclusion criteria for all groups were immunocompromised, initiation of antibiotics more than 48 h before, and age below 10 years. For the LD-positive group, exclusions extended to individuals with reactions to tick bites that did not result in EM or expanding annular lesions and those with a history of chronic fatigue syndrome, rheumatologic diseases, or multiple sclerosis. The LDB samples were characterized using screening ELISAs such as whole-cell sonicates, C6 peptide ELISA, or VlsE/pepC10, and IgM and IgG western blots were run regardless of the first-tier results[31]. Samples were classified as either Lyme positive or negative using STTT, and all 35 lab-confirmed samples were STTT positive, while all endemic controls were STTT negative. All samples were tested using PCR for *B. burgdorferi*, *B. miyamotoi*, *A. phagocytophilum*, and *B. microti* apart from the two-tier serology tests.

Sample cohort 4 was used for blinded evaluation of the xVFA platform and included 32 samples acquired from the CDC's Lyme serum repository research panel I, which consisted of 12 Lyme disease patient samples, eight healthy control samples collected from regions where Lyme disease is endemic and non-endemic, and 12 look-alike disease samples which are known to be cross-reactive with LD diagnosis[32]. The CDC LD repository included early acute LD-positive samples with localized EM lesions of at least 5 cm. For early disseminated and late-stage LD samples, clinical manifestations were consistent with each condition: varying degrees of heart block for Lyme carditis, specific neurological symptoms for neuroborreliosis, and intermittent or chronic oligoarticular arthritis for late Lyme arthritis. The diagnosis was corroborated wherever possible using

culture and/or PCR evidence of *B. burgdorferi* infection. Patients were excluded if they had significant immunocompromising conditions or if they were unable or unwilling to follow the protocol for sample collection. Additionally, for late Lyme arthritis, patients with a history of certain rheumatological conditions or syphilis were also excluded. All individuals were at least 18 years of age and had appropriate epidemiological risk at the time of collection. Detailed demographic information and sample-specific characterization for these samples are not included in this paper following the CDC's guidance. Further information regarding the CDC's LD samples repository can be found in a previous publication describing the repository development[32]. The CDC LDR samples were classified using both STTT and MTTT. For STTT, the panels were measured using VIDAS Lyme IgM and IgG polyvalent assay by bioMérieux for first-tier serology and MarDx Diagnostics, Inc IgM, and IgG immunoblotting assay for second-tier serology. For the MTTT classification, tests from Zeus VlsE/pepC10, Zeus WCS EIA were used as first-tier tests and Zeus IgM EIS and IgG EIA were used for second-tier serology. No training was performed on the CDC dataset.

### Statistics and reproducibility

The samples in this study were acquired based on the availability and characterization of the serum using two-tier serology. No sample size calculations were performed. Each of the samples were tested through three separate xVFA tests, and no data were excluded. The study includes samples from different biobanks to ensure the performance of the test across different cohorts comprising early disease samples and look-alike disease samples previously untested using the diagnostic algorithm. The samples from both the Bay Area Lyme Disease Biobank (LDB) and the CDC were received with deidentified and blinded labels. The training and validation cohort samples from LDB were tested using the developed xVFA assay, and subsequently, the results were shared with the Lyme Disease Biobank to reveal the disease classifications. The xVFA predictions for the CDC samples were shared with the CDC after analysis using the trained algorithm to reveal the sample labels.

### Reporting summary

Further information on research design is available in the Nature Portfolio Reporting Summary linked to this article.

## Data availability

All experiment data and supporting findings are presented in the paper and Supplementary Information in graphic and table form. Source data are provided with this paper.

## Code availability

The code for the machine learning algorithm used in this study is available from the corresponding author upon request due to pending intellectual property filings that prevent public disclosure. Access requests are subject to a signed Material Transfer Agreement (MTA), with requests for commercial use subject to licensing terms and conditions. Requests will be processed within one week after contacting the corresponding authors.

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

## Acknowledgements

The authors acknowledge the Bay Area Lyme Foundation's Lyme Disease Biobank and the US Centers for Disease Control & Prevention for graciously providing serum samples that were used for training and blinded validation of our platform respectively. We thank Wesley Luk for assistance with the preparation of xVFA tests. The work was funded by National Institute of health (Grant #R44AI150060) awarded to P.M.A. and D.D.C., and the National Science Foundation PATHS-UP Engineering Research Center (Grant #1648451) awarded to A.O. and D.D.C. Selected schematic figures were created using BioRender.com.

## Author contributions

R.G., H.-A.J., P.M.A., R.J.D., A.O., and D.D.C. conceptualized and developed the idea for the study. R.G., H.-A.J., E.J.H., O.B.G., P.M.A., A.O., and D.D.C. developed the methodology. R.G., H.-A.J., A.G., B.P., K.N., K.P., N.K., E.G., and A.O.K. collected and analyzed the data. R.G. and D.D.C. prepared the original draft of the manuscript. R.G., H.-A.J., E.J.H., P.M.A., R.J.D., A.O., and D.D.C. reviewed and edited the manuscript. P.M.A., R.J.D., A.O., and D.D.C. acquired funding for the study. A.O. and D.D.C. supervised the research. All authors reviewed and approved the final manuscript.

## Competing interests

H.-A.J., O.B.G., A.O., and D.D.C. are inventors in patents (US12013395B2) and patent applications (US20220299525A1) for the xVFA and smartphone reader platform. Some of the peptides described in this study are protected under US patent number 7887815B2 and US provisional patent application nos. 14376409 and 15102002, all owned by Biopeptides, Corp. R.J.D. is a shareholder in Biopeptides, Corp. P.M.A. has a research appointment with Biopeptides Corp. The remaining authors declare no competing interests.
