## [Peer Review File · Nature Communications]

REVIEWER COMMENTS

Reviewer #1 (Remarks to the Author):

Review of "Single-tier point-of-care serodiagnosis of Lyme disease," for submission to Nature Communications, by Ghosh et al.

The authors present a point of care serology test for Lyme Disease using synthetic peptides as the binding affinity agent for IgG and IgM. The developed assay addresses a shortcoming of the two tiered approach for serology for Lyme, which requires first ELISA and then a Western blot. This approach causes long turnaround times, and WB has some shortcomings in that it is qualitative.

They design and construct a multiplexed vertical flow assay (xVFA) and use ML to interpret the results of the resulting assay image. They first screen and select a set of antigenic peptides for *B. burgdorferi*, as synthetic peptides are cheaper, easier to make than recombinant proteins. To train the signal they use a deep learning neural network, and in the final device run patient serum samples in it and demonstrate high sensitivity and specificity. They also construct a smartphone based reader for readout of the fluorescent signals, using an LED for the excitation, so the overall device has a low cost.

Results are significant as there are not many existing Lyme disease diagnostics and diagnostic routes are suboptimal. The work presented here would be of interest to a broad audience and also the readership of Nature Communications.

There are some issues to be addressed before publication.

1) The assay does not perform so well with early acute infections (Figure 4a). This makes sense as the immune response has not yet produced enough antibodies for detection. However, it would be good if this were corroborated by measuring antibody titers of the samples.

2) What is used as the positive control in Figure 3b? What is the rationale for this choice?

3) There are some areas in which details of what is being presented in the figures would be helpful. For example, how the intensities in Figure 2b,c, Figure 3b are measured should be described in the figure caption and text.

4) Some citations incorrect and the manuscript should be proofread with respect to which citations are cited where.

5) In Figure 2, there is no panel labeled for Figure 2c.

Reviewer #2 (Remarks to the Author):

This is a well researched, well written manuscript. The authors describe a synthetic peptide based, machine learning strategy to refine predictive value in Lyme disease that might be applicable to a variety of infectious diseases. Their approach is efficient, simple, cost effective. They document that in comparison to a small population of reference samples, their final peptide test is non-inferior to gold standard 2 tier testing.

They stress cost, convenience and ease of this testing. I agree that this is an exciting, worth while methodology. I am less convinced that they address unmet needs in Lyme disease testing. First early Lyme disease diagnosis is primarily clinical since sensitivity, even with reliable testing is low. Second, false positives are common because pre-test probability is low in the vast number of Lyme disease tests performed in the USA annually.

For these reasons, the goal should be a better test, not a more available test. And while the authors claim non inferiority, their test is less sensitive in the important acute infection cohort (Figure 4B).

Reviewer #3 (Remarks to the Author):

The authors present the development and evaluation of a Point-of-Care test (POCT) using synthetic peptides to detect IgG and IgM antibodies to *B. burgdorferi* in a single step assay. The assay uses a vertical flow (xVFA) cartridge format that the authors have previously developed that offers rapid results and claims low-cost. The multiplex assay using 3 different peptides, modVlsE-FlaB, Var2FlaB and OppA4 showed comparable performance to two-tier testing (2TT) done in a central laboratory. Similar to 2TT, this single tier POCT did not detect antibodies in several patients with

acute EM using the CDC serum panel, although modified 2TT showed a better performance than xVFA on these samples. One of the goals of newly developed assays, that these authors were aiming at, is to increase sensitivity in early disease that was not achieved in this study.

Based on the complexity of serodiagnosis of Lyme disease, a POCT that has similar performance characteristics to the currently available assays and that potentially could be run by non-laboratory professionals does not seem to be ready for diagnostic use. Perhaps further evaluations of this xVFA using a larger well-characterized sample set could confirm its improved performance over currently available tests. In addition, whether this test would be approved by FDA as a waived or moderate complexity needs to be determined.

Specific comments

Line 11: As it stands now, POC serological testing is not likely to help on “difficult to diagnose illnesses” such as Lyme where the pathogen antigen complexity and the development of antibodies is variable. POC tests (POCT) for serological diagnosis have been useful for illnesses where there is clear knowledge of the pathogen immunodominant antigens and temporal appearance of the antibodies that they elicit. An example of this use is for the diagnosis of EBV infections. Perhaps the authors could clarify if they are referring to antigen detection using POCT. Certainly, POCT has been of most use to detect antigens directly on samples from difficult to diagnose diseases such as viral infections.

Line 38: The authors should emphasize that early Lyme disease presents frequently with Erythema migrans (EM) and the diagnosis at this stage is clinical. Serological testing is only recommended for patients presenting with atypical skin lesions or with extracutaneous manifestations of Lyme disease. They could refer here to Lantos et al. 2020 Guideline on Prevention, Diagnosis and Management of Lyme disease *Arthritis Rheumatol.* 2021 Jan;73(1):12-20.

Line 43: Suggest adding “In the absence of EM” the serological (antibody) testing of LD.....

Lines 90-92: Has the shelf-life of the combined synthetic peptides been evaluated? This is an important aspect of quality control of POCT assays that would be used outside a central laboratory.

Line 130: Figure 2c is missing the label “c”.

Lines 173-177: a clarification is needed of which samples from the LDB were used to optimize the architecture of the neural network model as well as to validate the optimized model. Which cohort of samples do LD50, 72 and 74 belong to? LDB samples are labeled LDB1 – 50 on supplementary Table 3.

Figure 4a. I suggest using a different and distinct color for the dots representing Lyme (acute, conv and late stage) results. It is difficult to distinguish between the different shades of pink/red.

Line 260-264: As mentioned before, are the authors considering this xVFA test as a waived or moderate complexity test? Only if FDA considers this test as waived could be used at localities outside of a central laboratory.

Reviewer #4 (Remarks to the Author):

Complete comments as follows:

Summary: The authors present a diagnostic platform, the multiplexed vertical flow device (xVFA), designed for the rapid and cost-effective detection of Lyme disease. This study builds upon their previous works (Joung et al., 2019 and 2020, see below) with the main distinction being the use of synthetic peptides in place of recombinant proteins, offering advantages including increased shelf-life and cost-effectiveness. A set of epitopes were selected and trained by machine learning to yield the final diagnostic results. Although the test has demonstrated advantages over existing methods in the ease of implementation, the major concern would be as to how clear the advancements are when compared to the authors previously published work.

The paper could be published if the points are addressed carefully

Major Points

1. Significant advancement compared to previous studies?

The authors have already published similar approaches in their two previous works published by the same two lead authors (ACS Nano 2020, 14 (1), 229–240. <https://doi.org/10.1021/acsnano.9b08151>, Lab. Chip 2019, 19 (6), 1027–1034. <https://doi.org/10.1039/C9LC00011A>) The difference compared to the previous approach was the use of synthetic peptides instead of recombinant proteins, the merits of which are increased are stated as being increased shelf-life and costs.

Evidence for increased shelf life is just provided in the legend to Figure 1f. If this a major reason for the publication of the paper this needs to be described in more detail with more substantive comparisons showing a like for like comparison of the devices.

Evidence for cost is less clear, as the ACS Nano paper 2020 work claimed \$0.42 per test for material costs and this work suggests \$3

2. Study design

There was little information on the study design and the description of the patients' demographics or clinical characteristics. Given that the authors have completed a Reporting Summary, these points should be clearly given.

It is actually nice for the authors to include samples from multiple sources and centres for better validation. However, it is important to note the much larger differences in baselines of samples coming from multicentres. Have the authors look at the demographics of the patients and confounding variables that might influence the outcome of the diagnostic test? Were there co-morbidities, other medicines, etc? A diagnostic test cannot be accurately assessed without knowing the cohort baseline characteristics.

3. Performance evaluation

Except for the specificity, actual numbers for the performance of the test on clinical samples were not given in the results. From Figure 4b, sensitivity including early acute seems quite low. If this is the case the authors should comment on the low sensitivity of the test, especially for early onsets, compared to the already available methods. The lower sensitivity means less suitability as a screening test. To emphasise this point the performance of the new test, described here appears to be not as good as the previously published tests AUC 0.95, sensitivity 0.91, and specificity 0.87 (the authors do not currently show AUC and sensitivity numbers in this study).

To directly compare the performance, it would be useful to provide a single figure with a ROC curve, AUC, sensitivity and specificity for the three tests (this and the two previous publications).

To clearly show the efficacy of the test compared to existing method the authors should compare the new test and the reference test with a Bland and Altman plot.

Minor Points:

1. "OppA4 was given higher importance over other peptides". How was the importance assigned and translated into the design? Was the selection done manually? The initial selection process of 9 peptides needs more details and rationales.

2. The authors used 4 cohorts. It was confusing in the text where some cohorts were referred by number and some by abbreviations. The authors should generate a concise summary figure/table to show the characteristics, number of patients and their usage in the study, e.g. for peptide screening, machine learning training/testing, etc.

3. Figure 1g, shows large error bar and no statistics.

4. I think that Figure 2c-e should be Figure 2b-d. Also note capitalisation of figure names is not consistent.

5. The IRB approval document numbers for the three centres should be quoted.

Response to Reviewer Comments

Reviewer #1 (Remarks to the Author):

Review of "Single-tier point-of-care serodiagnosis of Lyme disease," for submission to Nature Communications, by Ghosh et al.

The authors present a point of care serology test for Lyme Disease using synthetic peptides as the binding affinity agent for IgG and IgM. The developed assay addresses a shortcoming of the two tiered approach for serology for Lyme, which requires first ELISA and then a Western blot. This approach causes long turnaround times, and WB has some shortcomings in that it is qualitative.

They design and construct a multiplexed vertical flow assay (xVFA) and use ML to interpret the results of the resulting assay image. They first screen and select a set of antigenic peptides for *B. burgdorferi*, as synthetic peptides are cheaper, easier to make than recombinant proteins. To train the signal they use a deep learning neural network, and in the final device run patient serum samples in it and demonstrate high sensitivity and specificity. They also construct a smartphone based reader for readout of the fluorescent signals, using an LED for the excitation, so the overall device has a low cost.

Results are significant as there are not many existing Lyme disease diagnostics and diagnostic routes are suboptimal. The work presented here would be of interest to a broad audience and also the readership of Nature Communications.

Response: We thank the reviewer for their careful evaluation of the manuscript and highlighting the significance of the work.

There are some issues to be addressed before publication.

1) The assay does not perform so well with early acute infections (Figure 4a). This makes sense as the immune response has not yet produced enough antibodies for detection. However, it would be good if this were corroborated by measuring antibody titers of the samples.

Response: We agree that determining antibody titers for the undetected acute Lyme Disease (LD) samples would be insightful. The acute LD samples undetected in the CDC panel had OD values for the different EIA tests that are significantly lower than the respective Cut-off Threshold (Ct) values. Notably, none of the samples were detected using the VIDAS Lyme IgM and IgG polyvalent assay. Only 1 of the 4 samples was detected using the WCS ELISA assay, 1 out of 4 by the Zeus IgM ELISA, and 2 of the 4 by the Zeus VlsE/PepC10 ELISA. It's important to note that the samples testing positive were marginally above the threshold, and their OD values were substantially lower compared to those from convalescent phase LD samples or other Late Lyme cases, indicating low antibody titer against LD antigens. While we can discuss the OD values qualitatively, unfortunately, specific OD values are unable to be reported in the manuscript given the CDC's guidelines for use of their repository under a materials transfer agreement.

The following sentence is added to the methods section of the sample description for the CDC cohort to highlight this aspect, "Detailed demographic information and sample specific characterization for these samples are not included in this paper following the CDC's guidance. Further information regarding the CDC's LD samples repository can be found in a previous publication describing the repository development³²."

2) What is used as the positive control in Figure 3b? What is the rationale for this choice?

Response: Goat-anti-mouse was used as positive control for the xVFA test to validate the binding of the Mouse anti-human antibodies conjugated to gold nanoparticles on the sensing membrane surface. The positive control spots were included to ensure that the conjugated gold nanoparticles were functional, flowing through the xVFA paper layers, and stably bound to the surface of the sensing membrane.

We revised the sentence in the methods section of the revised manuscript to clarify this rationale, "... to ensure that the gold nanoparticle conjugated mouse anti-human secondary antibodies successfully flowed through the xVFA and bound to the sensing membrane."

3) There are some areas in which details of what is being presented in the figures would be helpful. For example, how the intensities in Figure 2b,c, Figure 3b are measured should be described in the figure caption and text.

Response: The plots presented in figure 2b represent a heatmap of the normalized signal intensities obtained after performing the xVFA assay with six positive LD samples and six healthy control samples. The normalized signal intensity is obtained by dividing the post-assay signal with the pre-assay background and subtracting the fraction from 1. Figure 2c presents the variance of the normalized signal intensity across the patient samples tested and is plotted in the order of decreasing variance to indicate the reactivity of antigen peptides and highlight the most reactive ones. Figure 3b, presents the average normalized intensity of the individual peptide-based immunoreactions on the xVFA against the clinical patient samples obtained from the Lyme Disease Biobank. The manuscript has been amended in the figure text description, figure caption and methods to reflect this clearly:

Page 4, second paragraph reads, "...Figure 2b presents a heatmap illustrating the normalized signal intensities for each peptide spot, as measured from the assay results. These intensities correspond to the signals detected and captured in images recorded by the smartphone-based reader..."

"...Figure 2c illustrates the variance in normalized signal intensities across both LD positive and negative samples on the sensing membrane, effectively representing the activity of the peptides..."

Page 5, first paragraph reads, "...Figure 3b illustrates that the average of mean normalized signal intensities, calculated from two immunoreaction spots across three independent assay replicates of xVFA tests performed on a single patient sample..."

Under Methods section, page 8, first paragraph reads, "...The normalized signal intensity is calculated by dividing the post-assay signal by the pre-assay background, and then subtracting this fraction from 1. This normalizes the raw intensities of the reaction spots against their pre-assay background. Subsequently, the resulting normalized signal intensity is analyzed through a machine learning based multiplexed diagnostic algorithm to determine final prediction values and seropositivity of the tested sample..."

Figure 2 caption has been amended and reads, "... **b** Heatmap representing the normalized signal intensities of peptides screened against patient samples positive (+) and negative (-) for LD, utilized in the development of the paper-based multi-antigen xVFA platform. **c** Variance in the normalized signal intensity for each antigen peptide, determined through xVFA screening of control patient sera. The variance is plotted in descending order of reactivity, illustrating the comparative activity of the peptides against both positive and negative patient serum samples, highlighting the most active antigens..."

Figure 3 caption reads, "...b Heatmap displaying the average normalized signal intensities (N=3) derived from the multi-antigen panel, illustrating the results for each patient sample tested from the LDB against individual antigen peptides along with positive and negative control spots..."

4) Some citations incorrect and the manuscript should be proofread with respect to which citations are cited where.

Response: Thank you for the careful reading. References 14 and 15 (11 and 12 of the old manuscript) have been changed in the revised manuscript to be cited at end of the sentence. We did not find any other errors.

5) In Figure 2, there is no panel labeled for Figure 2c.

Response: The figure has been corrected to include the missing label.

Reviewer #2 (Remarks to the Author):

This is a well researched, well written manuscript. The authors describe a synthetic peptide based, machine learning strategy to refine predictive value in Lyme disease that might be applicable to a variety of infectious diseases. Their approach is efficient, simple, cost effective. They document that in comparison to a small population of reference samples, their final peptide test is non-inferior to gold standard 2 tier testing.

Response: We thank the reviewer for careful evaluation of the manuscript and the constructive feedback.

They stress cost, convenience and ease of this testing. I agree that this is an exciting, worth while methodology. I am less convinced that they address unmet needs in Lyme disease testing. First early Lyme disease diagnosis is primarily clinical since sensitivity, even with reliable testing is low. Second, false positives are common because pre-test probability is low in the vast number of Lyme disease tests preformed in the USA annually.

For these reasons, the goal should be a better test, not a more available test. And while the authors claim non inferiority, their test is less sensitive in the important acute infection cohort (Figure 4B).

Response: We agree with the reviewer that the goal should be a better test and not just a more available test. We also agree that the diagnosis of early LD comprises clinical evaluation of symptoms or presence of EM. However, laboratory confirmation of LD via the CDC's recommended two-tier serology continues to remain a mainstay for diagnosis confirmation. A 2008 survey of seven large U.S. commercial laboratories reported to have conducted about 3.4 million tests, revealing approximately 288,000 infections among the 2.4 million tested patients. With the number of LD cases rising each year the number of laboratory tests carried out for Lyme disease is only expected to increase, thus demonstrating the need for expanded testing.

In the introduction of the revised manuscript, we highlight, "...Early LD diagnosis typically relies on clinical evaluation, essentially the presence of erythema migrans (EM) skin lesions. However, laboratory confirmation is essential in cases with atypical rashes or extracutaneous signs or symptoms, as these symptoms often overlap with those of other illnesses^{8,9}. Early detection and treatment are crucial to prevent the dissemination of bacteria to a variety of distal sites, resulting in serious tissue-specific manifestations, including neurological, cardiac, or rheumatoid complications¹⁰. The current standard for

laboratory diagnosis is a two-tier testing process conducted in centralized facilities, primarily due to the lack of tests with sufficient specificity for a single-tier approach. With over 3.4 million LD tests conducted each year¹¹, there is a critical need for a single-tier POC test that can facilitate rapid diagnosis and treatment¹².”

In our revised manuscript, we also conduct additional testing of early acute LD samples, and our data provides further evidence of the xVFA's enhanced performance in detecting early acute LD samples from the LDB cohort, particularly those that had not undergone seroconversion, indicating an early onset of the disease. In our initial submission, the training cohort included 25 LD-positive samples out of which 15 were acute stage samples that had not undergone seroconversion. Additionally, in the revised manuscript, we included an additional testing of 10 early acute samples in the testing panel of the previously developed machine learning model out of which 7 were acute stage samples without seroconversion. Out of the total 22 early acute LD samples in the LDB cohort that had not undergone seroconversion, 20 were successfully detected by the xVFA, with only two early acute LD samples remaining undetected. This equates to a 90.9% sensitivity of the xVFA in detecting early acute LD samples, demonstrating its effectiveness in early disease diagnosis. We have included these results in the performance evaluation of our trained diagnostic model, as illustrated in the updated Figure 3e. The samples that did not undergo seroconversion are highlighted in blue in Figure 3e for easy visualization.

The following changes are made in the results section of the revised manuscript to address these concerns,

“To train the deep-learning diagnostic model, a cohort of 60 serum samples (See methods, Sample cohort 3) was used. The cohort included 35 early-stage LD samples and 25 healthy control samples collected by the Lyme disease biobank (LDB) from LD endemic regions (Supplementary Table 3). Of these, 22 samples were from patients with acute stage LD infection whose serum did not show signs of seroconversion and were positive only for the IgM antibodies when tested using standard tests. Further, the pool consisted of patients who reported experiencing a tick bite with timeframes ranging from one to four weeks from the tick bite to the time of blood draw, thus ensuring that early onset LD disease samples were represented and used in the training and testing of the developed assay...”

“...Among the 22 early acute LD samples from the LDB cohort, the xVFA successfully detected 20 samples, with the exception of only two samples, which demonstrates a sensitivity of 90.9% in detecting acute infections and underscores the xVFA's capability in recognizing early stages of LD...”

In our revision, we also note that while traditional laboratory tests employ a two-tier diagnostic approach, requiring a patient to test positive on both tests for a confirmed LD diagnosis, a single-tier testing approach, as demonstrated in this work, offers potential solutions that mitigate the challenges associated with two-tier testing. The sequential nature of two-tier serology can lead to compounded inaccuracies and poor diagnostic outcomes, such as reduced sensitivity due to the necessity of multiple positive test results and false positives outcomes. For example, the low specificity of the first-tier tests used in STTT and MTTT could result in significant false positives, thereby increasing the testing burden for second-tier tests. Similarly, the low sensitivities of the individual first- and second-tier tests could potentially lead to a large fraction of false negative results when tested across a large number of patient samples. These limitations and the potential mitigation offered by a single-tier POC test are now clearly discussed in the manuscript to highlight the advances of the xVFA over conventional two-tier serology testing.

Based on the reviewer comments, the following revisions are made in the results section of the revised manuscript, “Overall, our POC xVFA achieved a sensitivity of 61% and specificity of 100% (average performance of triplicate testing) which closely matches the performance of the standard two-tier testing algorithm (Figure 4). Figure 4b shows the prediction values of the trained machine learning diagnostic

model for the blinded validation panel. Both the single-tier xVFA and the STTT did not detect any of the acute LD samples in the panel. This could be attributed to the low LD-specific antibody levels in this particular cohort of early acute LD samples, requiring follow-up testing until a detectable immune response can be measured using these tests. Only the modified two-tier testing algorithm performed slightly better using the IgM based diagnosis and was able to classify one additional acute LD sample as LD positive resulting in a sensitivity of 75%. As represented in Figure 4b and 4c, when comparing the performance of the single-tier xVFA assay with the individual centralized lab assays, we found that the xVFA performs equally or better than many of the individual laboratory assays. For example, although sensitive towards some of the acute LD samples, the modified two-tier tests were also positive for some of the non-LD control samples resulting in a specificity of 80% in some cases. This would be deemed unacceptable for a single-tier assay. It is only when the two-tier test assays are combined that the specificity is improved, highlighting the potential for erroneous reporting of the individual assays, and the current necessity for the two-tier algorithms....”

“The performance of the xVFA closely mirrored that of the standard two-tier testing algorithms, identifying the same patient samples as either positive or negative. Moreover, the xVFA offers the added benefits of being a single-tier test and having a rapid POC format that could be interpreted during clinical evaluation of patients...”

The following discussions are added in the discussion section of the revised manuscript “The standard and modified two-tier serology based diagnostic algorithms comprise individual tests that have varying sensitivities and specificities that may lead to compounded inaccuracies in diagnosing LD. For instance, within the standard two-tier framework, the VIDAS Lyme IgM/IgG, IgM Western blot, and IgG Western blot tests have sensitivities of 67%, 58%, and 42%, respectively. The low sensitivity of the tests, combined with the tiered approach of testing can potentially lead to a decline in overall sensitivity, increasing likelihood of missing true positive cases when tested across a larger patient cohort. Furthermore, the first tier VIDAS Lyme IgM/IgG assay's specificity of 80% raises the probability of false positives, necessitating additional confirmatory testing and thus escalating the burden, cost, and complexity of LD testing. Similarly, the modified two-tier framework comprises tests with similar limitations with sensitivity of the first-tier test being 83% and the second-tier IgM and IgG EIA being 75% and 42% respectively. Additionally, the VlsE/PepC10 EIA in the first tier of the MTTT exhibits a specificity of 95%, but the specificity drops to 80% in the second-tier IgM EIA. This reduction in specificity can significantly increase the rate of false positive diagnoses, adversely impacting the positive predictive value of the modified two-tier framework. Conversely, a single-tier assay demonstrating a sensitivity ~61%, coupled with a specificity of 100%, can overcome the limitations posed by sequential testing, thereby offering significant cost-saving with rapid turnaround time and improved diagnostic outcomes. Unlike the two-tier approach, where the low specificity of the first-tier tests necessitates additional follow-up tests and management of false positive patients, a single-tier assay could reduce the complexity of diagnosis offering rapid disease confirmation, potentially leading to cost savings and timely treatment.”

With these revisions based on the reviewer feedback, we believe that we have addressed the concerns raised by the reviewer and help highlight the significance of the work with regards to LD testing.

Reviewer #3 (Remarks to the Author):

The authors present the development and evaluation of a Point-of-Care test (POCT) using synthetic peptides to detect IgG and IgM antibodies to *B. burgdorferi* in a single step assay. The assay uses a

vertical flow (xVFA) cartridge format that the authors have previously developed that offers rapid results and claims low-cost. The multiplex assay using 3 different peptides, modVlsE-FlaB, Var2FlaB and OppA4 showed comparable performance to two-tier testing (2TT) done in a central laboratory. Similar to 2TT, this single tier POCT did not detect antibodies in several patients with acute EM using the CDC serum panel, although modified 2TT showed a better performance than xVFA on these samples. One of the goals of newly developed assays, that these authors were aiming at, is to increase sensitivity in early disease that was not achieved in this study.

Based on the complexity of serodiagnosis of Lyme disease, a POCT that has similar performance characteristics to the currently available assays and that potentially could be run by non-laboratory professionals does not seem to be ready for diagnostic use. Perhaps further evaluations of this xVFA using a larger well-characterized sample set could confirm its improved performance over currently available tests. In addition, whether this test would be approved by FDA as a waived or moderate complexity needs to be determined.

Response: We thank the reviewer for careful evaluation of the manuscript and raising pertinent concerns that aim to improve the quality of the work.

Regarding the xVFA test's efficacy in detecting early-stage Lyme disease (LD), we highlight in our revised manuscript that the xVFA was both trained and evaluated (tested) using numerous early-stage samples from the Bay Area Lyme Disease Biobank (LDB). Of the initial 25 LD-positive samples, 15 were acute-stage samples that had not undergone seroconversion. Additionally, based on the reviewer feedback we conducted further testing with 10 early-stage LD samples, of which 7 had not undergone seroconversion. Out of the 22 early acute LD samples in the LDB cohort that had not undergone seroconversion, 20 were successfully detected by the xVFA, with only two early acute LD samples remaining undetected. This equates to a 90.9% sensitivity of the xVFA in detecting early acute LD samples, demonstrating its effectiveness in early disease diagnosis. We have included these results in the performance evaluation of our trained diagnostic model, as illustrated in the updated Figure 3e. The samples that did not undergo seroconversion are highlighted in Figure 3e by color coding for easy visualization.

The following revisions are made in the revised manuscript under the results section addressing these concerns: “To train the deep-learning diagnostic model, a cohort of 60 serum samples (See methods, Sample cohort 3) was used. The cohort included 35 early-stage LD samples and 25 healthy control samples collected by the Lyme disease biobank (LDB) from LD endemic regions (Supplementary Table 3). Of these, 22 samples were from patients with acute stage LD infection whose serum did not show signs of seroconversion and were positive only for the IgM antibodies when tested using standard tests. Further, the pool consisted of patients who reported experiencing a tick bite with timeframes ranging from one to four weeks from the tick bite to the time of blood draw, thus ensuring that early onset LD disease samples were represented and used in the training and testing of the developed assay...”

“...Among the 22 early acute LD samples from the LDB cohort, the xVFA successfully detected 20 samples, with the exception of only two samples, which demonstrates a sensitivity of 90.9% in detecting acute infections and underscores the xVFA's capability in recognizing early stages of LD...”

Regarding the reviewer comments on performance of the xVFA in the CDC panel, we agree that the acute LD samples from this specific CDC cohort were undetected using the xVFA test. This is likely attributable to early sample collection and the absence of a detectable immune response in these patients against the pathogen, considering that LD-specific IgM antibodies typically take around 2-4 weeks to form post-tick bite, followed by IgG production. The low antibody levels in these samples are

corroborated by the inability of the VIDAS Lyme IgM/IgG to detect any of these acute stage samples, as shown in Supplementary Table 4 and Figure 4. Only two patient samples tested positive with the Zeus VlsE/pepC10 EIA, and one patient sample tested positive using the standard IgM Western blot, Zeus WCS EIA, and Zeus IgM EIA. Consequently, none of these acute LD samples were classified as positive using the standard two-tier serology, and only one sample was classified as positive using the modified two-tier algorithm. Given that serological testing is the primary approach recommended by the CDC, these tests will not yield satisfactory results in the absence of a detectable immune response in patients post-tick exposure. In such scenarios, repeat testing is advised, with blood draws occurring several days later during the convalescent stage to check for any immune response, thus aiding in the laboratory confirmation of the disease.

Additionally, we also note that the modified two-tier EIAs which were positive for some acute samples, also yielded positive results for a cross-reactive sample from a patient with Syphilis and three healthy control samples from both endemic and non-endemic LD regions. Despite the individual tests of the MTTT showing sensitivity for the acute samples in the CDC panel, the individual assays' specificities were relatively low, with the first-tier VlsE/PepC10 EIA at 95% and the second-tier IgM EIA at 80%, demonstrating a high risk of false positive outcomes and poor positive predictive value. Figure 4 in the revised manuscript is updated to indicate the individual performances of the first and second-tier assays as compared to the xVFA.

The following changes are made in the results section of the revised manuscript to address the concerns:

“Overall, our POC xVFA achieved a sensitivity of 61% and specificity of 100% (average performance of triplicate testing) which closely matches the performance of the standard two-tier testing algorithm (Figure 4). Figure 4b shows the prediction values of the trained machine learning diagnostic model for the blinded validation panel. Both the single-tier xVFA and the STTT did not detect any of the acute LD samples in the panel. This could be attributed to the low LD-specific antibody levels in this particular cohort of early acute LD samples, requiring follow-up testing until a detectable immune response can be measured using these tests. Only the modified two-tier testing algorithm performed slightly better using the IgM based diagnosis and was able to classify one additional acute LD sample as LD positive resulting in a sensitivity of 75%. As represented in Figure 4b and 4c, when comparing the performance of the single-tier xVFA assay with the individual centralized lab assays, we found that the xVFA performs equally or better than many of the individual laboratory assays. For example, although sensitive towards some of the acute LD samples, the modified two-tier tests were also positive for some of the non-LD control samples resulting in a specificity of 80% in some cases. This would be deemed unacceptable for a single-tier assay. It is only when the two-tier test assays are combined that the specificity is improved, highlighting the potential for erroneous reporting of the individual assays, and the current necessity for the two-tier algorithms. The CDC panel samples were tested in triplicate using the xVFA. In repeat testing of the xVFA, one replicate failed to detect either a convalescent or late-stage LD sample, resulting in a slightly reduced sensitivity of 61% compared to the single replicate performance of the STTT.”

Regarding the concern about FDA CLIA-waived approval please see the specific comment below for our response.

Specific comments

Line 11: As it stands now, POC serological testing is not likely to help on “difficult to diagnose illnesses” such as Lyme where the pathogen antigen complexity and the development of antibodies is variable. POC tests (POCT) for serological diagnosis have been useful for illnesses where there is clear knowledge of

the pathogen immunodominant antigens and temporal appearance of the antibodies that they elicit. An example of this use is for the diagnosis of EBV infections. Perhaps the authors could clarify if they are referring to antigen detection using POCT. Certainly, POCT has been of most use to detect antigens directly on samples from difficult to diagnose diseases such as viral infections.

Response: The statement is revised to include POCT antigen testing and now reads, “Point-of-care (POC) serological and direct antigen testing provides actionable information for several difficult to diagnose illnesses, empowering distributed health systems.”

Line 38: The authors should emphasize that early Lyme disease presents frequently with Erythema migrans (EM) and the diagnosis at this stage is clinical. Serological testing is only recommended for patients presenting with atypical skin lesions or with extracutaneous manifestations of Lyme disease. They could refer here to Lantos et al. 2020 Guideline on Prevention, Diagnosis and Management of Lyme disease *Arthritis Rheumatol.* 2021 Jan;73(1):12-20.

Response: The manuscript has been updated to emphasize the clinical diagnosis of early Lyme disease, particularly in cases presenting with the characteristic erythema migrans (EM) rash.

The sentence in the introduction section in the revised manuscript now reads as “Early LD diagnosis typically relies on clinical evaluation, essentially the presence of erythema migrans (EM) skin lesions. However, laboratory confirmation is essential in cases with atypical rashes or extracutaneous signs or symptoms, as these symptoms often overlap with those of other illnesses^{8,9}.”

Line 43: Suggest adding “In the absence of EM” the serological (antibody) testing of LD.....

Response: The sentence is revised and reads, “In the absence of EM, serological (antibody) testing of LD...”

Lines 90-92: Has the shelf-life of the combined synthetic peptides been evaluated? This is an important aspect of quality control of POCT assays that would be used outside a central laboratory.

Response: In our initial submission, we assessed the performance of a single peptide and a protein antigen to compare their stability. In response to the reviewer's comments, we have carried out additional experiments to assess the shelf-life of the combined synthetic peptide panel and the trained machine learning diagnostic algorithm. The data measuring the antibody response of a patient with LD positive serum and a healthy control sample using the entire peptide panel over 60 days (limited by the revision resubmission deadline) are now presented in Supplementary figure S1 as supporting information. The output of the entire machine learning model that makes inferences based on the different immunoreaction intensities remains stable for both the LD positive and healthy control over this time. The methodology of the study is explained in the methods section under sub-section ‘Stability of antigens and shelf-life testing’. The statement reads as -

“Stability of antigens and shelf-life testing

To assess the stability of the peptide and protein antigen targets after immobilization on the nitrocellulose membrane, sensing membranes were spotted with 1 mg/mL of modVlsE-FlaB peptide and VlsE native *B. burgdorferi* protein and immobilized in three immunoreaction spots each. These membranes were then tested using control Lyme disease positive and healthy samples at various time points between days zero

and ninety. The signal intensities corresponding to the peptides and proteins were compared to determine the stability of the respective antigens. Further, shelf-life of the combined synthetic peptide panel, spotted on the nitrocellulose membrane, were tested over time to determine the consistency of performance of the full assay over 60 days of testing at various time points. Each immunoreaction spot was coated with either 1 mg/mL of the respective peptide, goat anti-mouse IgG as positive control or PBS buffer with 1% BSA as negative control as shown in Figure 3a. All xVFA cases for stability and shelf-life measurements were prepared and stored at 25°C until tested. The results from each individual immunoreaction spot were processed through the optimized machine learning diagnostic model, which yielded the prediction values for each xVFA test. These values were then used to assess the assay's consistency and reliability over the evaluation period.”

A discussion of these results is also included in the results section, “Additionally, the combined synthetic peptide panel demonstrated similar remarkable stability without significant loss in reactivity or signal over the course of 60 days of testing, as illustrated in Supplementary Figure S1.”

Line 130: Figure 2c is missing the label “c”.

Response: The figure label 2c has been corrected in the revised manuscript.

Lines 173-177: a clarification is needed of which samples from the LDB were used to optimize the architecture of the neural network model as well as to validate the optimized model. Which cohort of samples do LD50, 72 and 74 belong to? LDB samples are labeled LDB1 – 50 on supplementary Table 3.

Response: We have updated the supplementary table 3 to specify the samples used from cohort 3 for training and validating (testing) the neural network model.

Additionally, the LDB sample numbers LDB50, 72 and 74 were corrected in the revised manuscript to match with the sample numbers in the supplementary table 3. The sentence in the results section now reads, “Of the 3 samples (LD08, LD12, LD13) ...”

Figure 4a. I suggest using a different and distinct color for the dots representing Lyme (acute, conv and late stage) results. It is difficult to distinguish between the different shades of pink/red.

Response: The dot plot in Figure 4b (formerly Figure 4a in the original manuscript) has been revised with contrasting colors and grey dotted lines to differentiate clearly among acute, convalescent, and late-stage LD samples.

Line 260-264: As mentioned before, are the authors considering this xVFA test as a waived or moderate complexity test? Only if FDA considers this test as waived could be used at localities outside of a central laboratory.

Response: The operation of the xVFA is similar to other paper-based lateral flow tests available as rapid diagnostic kits on the market for at-home or in-clinic use. Many of these tests have achieved CLIA-waived status due to their simplicity and low risk of causing harm. For instance, the Sofia Lyme 2, a first-tier Lyme serology assay developed by Quidel Ortho, is a lateral flow test that has been granted CLIA-waived status. However, it is important to note that this assay is approved only as a first-tier test and must be complemented with a second-tier assay, typically conducted in a centralized laboratory, for conclusive

diagnostic determinations for Lyme disease. The xVFA, with an operational mechanism similar to the Sofia Lyme 2 and other similar LFA-type rapid tests, is designed for use by non-trained personnel. Consequently, we anticipate it will be possible to achieve a CLIA-waiver for the xVFA test. However, it is important to note that achieving regulatory approval would necessitate additional commercialization efforts and research involving larger cohorts, and multi-site and multi-user clinical testing to evaluate the platform's performance and ease-of-use, which extends beyond the scope of this manuscript. We are actively talking to stakeholders and seeking funding to advance in this direction.

Reviewer #4 (Remarks to the Author):

Complete comments as follows:

Summary: The authors present a diagnostic platform, the multiplexed vertical flow device (xVFA), designed for the rapid and cost-effective detection of Lyme disease. This study builds upon their previous works (Joung et al., 2019 and 2020, see below) with the main distinction being the use of synthetic peptides in place of recombinant proteins, offering advantages including increased shelf-life and cost-effectiveness. A set of epitopes were selected and trained by machine learning to yield the final diagnostic results. Although the test has demonstrated advantages over existing methods in the ease of implementation, the major concern would be as to how clear the advancements are when compared to the authors previously published work.

The paper could be published if the points are addressed carefully

Major Points

1. Significant advancement compared to previous studies?

The authors have already published similar approaches in their two previous works published by the same two lead authors (ACS Nano 2020, 14 (1), 229–240. <https://doi.org/10.1021/acsnano.9b08151>, Lab. Chip 2019, 19 (6), 1027–1034. <https://doi.org/10.1039/C9LC00011A>) The difference compared to the previous approach was the use of synthetic peptides instead of recombinant proteins, the merits of which are increased are stated as being increased shelf-life and costs.

Evidence for increased shelf life is just provided in the legend to Figure 1f. If this a major reason for the publication of the paper this needs to be described in more detail with more substantive comparisons showing a like for like comparison of the devices.

Evidence for cost is less clear, as the ACS Nano paper 2020 work claimed \$0.42 per test for material costs and this work suggests \$3

Response: In our current study, we report significant advancements over the previous publications from our group in this subject matter. The differences are as follows:

The Lab on a Chip paper (Lab. Chip 2019, 19 (6), 1027–1034) was our first introduction of a multiplexed vertical flow assay platform using paper-based diagnostic technology. In this publication, we describe the engineering principles and design of the paper-based immunoassay platform and demonstrate the use of the device in carrying out indirect ELISA by detection of LD specific antibodies against three native

Borrelia proteins. The three antigens do not comprehensively target all LD specific antibodies from a patient sample. Further, this publication does not aim to develop a comprehensive platform for diagnosis of LD but rather demonstrates the use of a multiplexed paper-based test for detection of LD antibodies in spiked buffer and three human serum samples that are IgM controls.

The ACS nano paper (ACS Nano 2020, 14 (1), 229–240) was our first attempt to develop a comprehensive rapid diagnostic assay for LD leveraging the multiplexed testing offered by the xVFA platform. In that study, we reported a two-tier POC xVFA assay for the separate detection of IgM and IgG antibodies specific to recombinant Borrelia proteins and a single peptide antigen in two separate xVFA assays. We carried out training and blinded evaluation of this two-tier assay along with the machine learning diagnostic algorithm, using 100 clinical samples that were obtained from the Bay Area Lyme disease Biobank. The assay achieved a sensitivity of 90.5% and specificity of 87% without threshold tuning. The shortcoming of the work was that it primarily relied on native protein antigens that were found to have significant cross-reactivity with non-LD samples, consequently resulting in a poor specificity. Further, this test required two separate xVFA assays that made the diagnostic algorithm challenging and resulted in a loss of sensitivity in each of the IgM and IgG specific assays when directly compared with individual standard two-tier tests.

The current work reports the development of a single-tier rapid POC assay using synthetic peptides for LD diagnosis, advancing upon our previous research. Custom synthetic peptides offer remarkable effectiveness in detecting LD-specific antibodies. They can be designed to retain sequences of major epitopes that target immunodominant regions, while simultaneously eliminating non-specific sequences that could otherwise lead to cross-reactive results. By using short linear peptide sequences representing major *B. burgdorferi* epitopes we have developed a multiplexed panel that not only offered high sensitivity but also improved specificity that surpasses the some of the individual first and second tier assays used for two-tier diagnosis. In addition to its cost-effectiveness, ease of synthesis, and extended shelf life, the peptide-based multiplexed panel developed in this study has resulted in an improved assay that demonstrated superior performance in a single-tier format when compared to individual traditional lab tests. Additionally, a vital aspect of this study involved comparing the assay's performance against both individual two-tier laboratory tests and the overall two-tier diagnosis algorithms currently in use, ensuring a direct comparison across identical patient samples. This approach was essential to accurately assess the true performance of our assay in relation to the current state-of-the-art. Therefore, the single-tier LD assay was blindly evaluated using samples from the CDC panel for a direct comparison of the xVFA performance with current laboratory tests in the same sample panel. Our xVFA multiplexed panel demonstrated improved specificity compared to some of the individual first and second-tier EIAs, which is attributed to the selection of the specific peptide based epitopes in our multiplexed panel. Furthermore, the sensitivity of our platform was on par with the standard two-tiered algorithm and only missed one early acute sample identified by the IgM-based modified two-tier algorithm in terms of diagnosis, as shown in Supplementary Table 4 and Figure 4 of the revised manuscript.

The following statement in our original submission highlights the limitation of using recombinant antigens for detection of LD antibodies, “In a previous study, we developed a two-tier POC assay using a combination of recombinant proteins and a single peptide that detected individual IgM and IgG levels in serum for detection of LD¹⁹. However, we observed that recombinant proteins suffer from poor specificity as they could potentially cross-react with other patient samples, particularly look-alike diseases or other common bacterial infections, leading to false-positive results in Lyme disease diagnosis¹⁸. Apart from potentially low cross-reactivity and a high specificity, synthetic peptides have additional advantages of

reduced cost and increased shelf-life when compared to full-length recombinant proteins of *B. burgdorferi* (Figure 1f) which are critical features in a POC test.”

Further, to help elucidate the differences between these studies, Table 1 in the manuscript is revised to highlight the advances of the current assay with respect to our previous ACS Nano reported assay (the only comprehensive LD diagnosis assay previously reported by our group) and the laboratory two-tier platforms in terms of cost, simplicity of single-tier testing, performance, and turnaround time.

In response to the comment about the cost of the xVFA platform, the ACS Nano paper reports a cost of \$0.42 for the paper materials alone, excluding protein antigens and other reagents. In Supplementary Table 5 of the current manuscript, we provide a comprehensive cost comparison, accounting for all necessary materials in the protein and peptide-based assay platforms. The manufacturing cost for a single protein-based assay is approximately \$13, while the peptide-based xVFA incurs a cost of only about \$3. It's important to note that the previous ACS Nano assay required two separate xVFA assays for IgM and IgG antibody detection, whereas the current methodology uses a single assay with a fabrication cost of \$3 manufactured in a research lab setting. However, this cost is subject to variations based on factors like time, manufacturing scale, and other considerations.

2. Study design

There was little information on the study design and the description of the patients' demographics or clinical characteristics. Given that the authors have completed a Reporting Summary, these points should be clearly given.

It is actually nice for the authors to include samples from multiple sources and centres for better validation. However, it is important to note the much larger differences in baselines of samples coming from multicentres. Have the authors look at the demographics of the patients and confounding variables that might influence the outcome of the diagnostic test? Were there co-morbidities, other medicines, etc? A diagnostic test cannot be accurately assessed without knowing the cohort baseline characteristics.

Response: We thank the reviewer for bringing this point to our concern. In the revised manuscript we provide more detailed information on the description and demographics of the clinical samples obtained from the Bay Area Lyme disease biobank and the CDC. However, we also note that detailed information on the sample characteristics and demographics of samples from the CDC are not able to be revealed in this publication as per the conditions imposed by the CDC in our materials transfer agreement. However, additional information such as sample inclusion and exclusion criteria for the different samples groups of the CDC panel are described in the publication *Journal of Clinical Microbiology*, 52, 10, 3755-3762 (2014) (10.1128/JCM.01409-14) and has been referenced in this manuscript.

The description of samples obtained from the Bay Area Lyme disease Biobank used for training and testing of the diagnostic model in this study were revised as follows in the methods section under clinical sample description: “Sample cohort 3 was used for training and testing of the machine learning based diagnostic model. The Cohort consisted of 60 samples from the LDB and was used for both training and testing of the deep-learning diagnostic model (Supplementary Table 3). The samples were collected from East Hampton, NY, between 2014 and 2019 (EH), and Marshfield, WI, from 2016 to 2019 (WI)³¹. Samples were collected through LDB sponsor protocol with Institutional review board (IRB) approval Advarra IRB protocol Pro00012408. Within the LDB sample cohort, there was a nearly even gender distribution, with 48.3% females and 51.3% males. The cohort included 35 two-tier positive Lyme disease samples with signs and symptoms of early Lyme disease. Patient samples in this group were

sourced from individuals in Lyme disease-endemic regions, with EM lesions diagnosed by a physician, irrespective of size. Participants also included those from endemic areas presenting with symptoms like headache, fatigue, fever, chills, or musculoskeletal pain, or with a history of suspected tick exposure. Additionally, 25 healthy control samples were collected from regions where Lyme disease is endemic, with at least two years of recorded residency in the region. The overarching exclusion criteria for all groups were immunocompromise, initiation of antibiotics more than 48 hours before, and age below 10 years. For the LD-positive group, exclusions extended to individuals with reactions to tick bites that did not result in EM or expanding annular lesions and those with a history of chronic fatigue syndrome, rheumatologic diseases, or multiple sclerosis. The LDB samples were characterized using screening ELISAs such as whole cell sonicates, C6 peptide ELISA, or VlsE/PepC10, and IgM and IgG western blots were run regardless of the first-tier results³¹. Samples were classified as either Lyme positive or negative using standard two-tier serology (STTT), and all 35 lab-confirmed samples were STTT positive while all endemic controls were STTT negative. All samples were tested using PCR for *B. burgdorferi*, *B. miyamotoi*, *Anaplasma*, *B. microti* apart from the two-tier serology tests.”

Further information of the characterization and processing of the samples from the LDB can be found in the following publication *Journal of Clinical Microbiology*, 58, 6, e00032-20 (2020) (10.1128/JCM.00032-20) and has been referenced in the manuscript.

The following sentence is added under the description of sample cohort 4 for the CDC panel: “...The CDC LD repository included early acute LD-positive samples with localized EM lesions of at least 5 cm. For early disseminated and late-stage LD samples, clinical manifestations were consistent with each condition: varying degrees of heart block for Lyme carditis, specific neurological symptoms for neuroborreliosis, and intermittent or chronic oligoarticular arthritis for late Lyme arthritis. Diagnosis was corroborated wherever possible using culture and/or PCR evidence of *B. burgdorferi* infection. Patients were excluded if they had significant immunocompromising conditions or if they were unable or unwilling to follow the protocol for sample collection. Additionally, for late Lyme arthritis, patients with a history of certain rheumatological conditions or syphilis were also excluded. All individuals were at least 18 years of age and had appropriate epidemiological risk at the time of collection. Detailed demographic information and sample specific characterization for these samples are not included in this paper following the CDC’s guidance. Further information regarding the CDC’s LD samples repository can be found in a previous publication describing the repository development³²...”

3. Performance evaluation

Except for the specificity, actual numbers for the performance of the test on clinical samples were not given in the results. From Figure 4b, sensitivity including early acute seems quite low. If this is the case the authors should comment on the low sensitivity of the test, especially for early onsets, compared to the already available methods. The lower sensitivity means less suitability as a screening test. To emphasise this point the performance of the new test, described here appears to be not as good as the previously published tests AUC 0.95, sensitivity 0.91, and specificity 0.87 (the authors do not currently show AUC and sensitivity numbers in this study).

To directly compare the performance, it would be useful to provide a single figure with a ROC curve, AUC, sensitivity and specificity for the three tests (this and the two previous publications).

To clearly show the efficacy of the test compared to existing method the authors should compare the new test and the reference test with a Bland and Altman plot.

Response: We thank the reviewer for the constructive comments, and we have made significant revisions to our manuscript to address the reviewer's concerns. In our revised manuscript, we have detailed the performance evaluation of the xVFA assay in comparison with both standard and modified two-tier testing methods for LD diagnosis in the CDC panel. The results section of the revised manuscript reads as, "Overall, our POC xVFA achieved a sensitivity of 61% and specificity of 100% (average performance of triplicate testing) which closely matches the performance of the standard two-tier testing algorithm (Figure 4). Figure 4b shows the prediction values of the trained machine learning diagnostic model for the blinded validation panel. Both the single-tier xVFA and the STTT did not detect any of the acute LD samples in the panel. This could be attributed to the low LD-specific antibody levels in this particular cohort of early acute LD samples, requiring follow-up testing until a detectable immune response can be measured using these tests. Only the modified two-tier testing algorithm performed slightly better using the IgM based diagnosis and was able to classify one additional acute LD sample as LD positive resulting in a sensitivity of 75%. As represented in Figure 4b and 4c, when comparing the performance of the single-tier xVFA assay with the individual centralized lab assays, we found that the xVFA performs equally or better than many of the individual laboratory assays. For example, although sensitive towards some of the acute LD samples, the modified two-tier tests were also positive for some of the non-LD control samples resulting in a specificity of 80% in some cases. This would be deemed unacceptable for a single-tier assay. It is only when the two-tier test assays are combined that the specificity is improved, highlighting the potential for erroneous reporting of the individual assays, and the current necessity for the two-tier algorithms..."

The discussion section is revised to include the following statements, "The xVFA demonstrated 100% agreement with the Standard Two-Tier Testing (STTT) algorithm and 88% agreement with the Modified Two-Tier Testing (MTTT) algorithm when evaluated using the CDC panel, a sample set previously unseen by the diagnostic model, thereby establishing the xVFA as a reliable single-tier LD testing platform suitable for rapid POC testing of LD..."

"... Zeus IgM EIA, WCS EIA and other Zeus VIsE/pepC10 EIA all detected one or two of the acute-stage samples but were also found to non-specifically bind to look-alike disease samples and healthy patient samples from endemic and non-endemic areas, reducing specificity and the positive predictive value of these centralized lab tests..."

These revisions highlight that the xVFA did not detect any of the acute stage LD samples in the CDC panel, which is likely due to the low antibody titers in these samples. We also highlight that although the MTTT IgM algorithm detected one additional acute LD sample and had a sensitivity of 75%, it also included results from the VIsE/PepC10 EIA (first tier) and the IgM EIA (second tier) which had a specificity of 95% and 80% respectively which demonstrates a significant potential of increased false positive outcomes when tested across large patient samples. The xVFA on the other hand did not exhibit any cross-reactivity with any of the known potential cross-reactive samples in the CDC panel, demonstrating a reliable single-tier diagnostic test for LD.

Further, we have updated Figure 4 in the revised manuscript to highlight the performance of the individual first- and second-tier assays in the standard and modified framework along with the overall diagnosis outcome of each algorithm with the single tier xVFA assay. This facilitates a detailed comparative analysis of the xVFA's performance, using the same sample group to provide a true comparison in performance of the assay with respect to current laboratory tests. Additionally, in the revised manuscript, we have updated Table 1 to compare the performance of various testing approaches, based on samples sourced from the same biobanks, ensuring similarity in composition for an accurate and fair assessment.

In response to the reviewer's suggestion for a performance comparison between our current assay and those previously published in ACS Nano and Lab on a Chip, we must clarify that such a direct comparison is not feasible and would not provide a reliable analysis. The Lab on a Chip publication did not include any clinical sample testing and was not designed for LD diagnosis. The ACS Nano publication marked our first report on an LD diagnosis test that separately detected the IgM and IgG antibodies from individual patient samples. However, a direct comparison of this previous assay's performance with that of the performance of the current assay in CDC panel is not reliable due to differing sample compositions. The CDC panel includes different sample compositions including look-alike disease samples that are known to cross-react with LD, leading to significant differences that undermine the validity of any direct performance comparison between these assays.

In our revised manuscript, we have therefore performed a direct one-to-one comparison of the xVFA's performance against other laboratory tests using the same sample set, offering a more accurate evaluation of the xVFA alongside conventional laboratory tests (Table 1, Figure 4b and 4c, Supplementary Table 4).

In response to the reviewer's suggestion on performing Bland and Altman analysis to compare the performance of the xVFA with the reference tests, we must clarify that the xVFA assay results are derived from a machine learning diagnostic algorithm, providing binary outcomes (LD positive or negative) ranging from 0 to 1 with 0.5 as the threshold, not the signal intensity corresponding to the concentration of a specific analyte. At the same time, the varying positivity thresholds and measurement scales used by each reference test for a specific analyte complicate the direct comparison of results using Bland Altman analysis. Additionally, the xVFA and the reference tests employ different antigen targets, leading to variations in LD-specific antibody recognition. This means they do not measure exactly the same biomarkers, rendering a Bland-Altman plot comparison unfeasible. However, based on the reviewer's feedback, we have included a bar plot comparison in our revised manuscript (Figure 4b and 4c). This comparison juxtaposes the sensitivity and specificity of the reference tests with that of the xVFA, offering a clear and direct analysis of the overall diagnostic efficacy of each test.

Minor Points:

1. "OppA4 was given higher importance over other peptides". How was the importance assigned and translated into the design? Was the selection done manually? The initial selection process of 9 peptides needs more details and rationales.

Response: We thank the reviewer for highlighting this point. In the revised manuscript, we have clarified this statement and the reasoning behind our peptide selection process. OppA4 was not given higher importance over all peptides but was selected within the multiplexed panel based on its reactivity with 4 out of the 6 LD positive control samples and the fact that the antibody recognition across the samples using OppA4 was uncorrelated with other screened peptides ($r^2 < 0.4$). This indicated that OppA4 was able to capture unique antibody signatures compared to the other peptides in the panel and was thus selected as an epitope within the multiplexed panel.

Peptides were selected in the multiplexed panel based on their reactivity with at least 2 of the 6 LD positive control samples tested. Further, we determined the variance of each peptide in their reactivity against the 12 control samples from cohort 2 and then ranked them based on the variance. Peptides that indicated general non-specific behavior with all human serum samples were not selected. Further, we determined the correlation of the peptides in identifying similar peptides from the samples tested and identified pairs of peptides that were highly correlated (similar performance) and selected the one that was

most important from immunogenic standpoint and critical in the pathogenesis of the disease. We have revised the manuscript to explain this rationale for better clarity.

The results section under sub-section epitope mapping and multiplexed panel design:

“...Further, OppA4 was selected in the multiplexed panel as it was reactive with 4 out of the 6 LD patient sera and was uncorrelated ($R^2 \leq 0.4$ to all other peptides) with other peptides, indicating unique antibody recognition.”

“Following the initial analysis of antibody binding to 18 peptides, we downselected to a subset of 9 peptides based on specific criteria. We selected peptides that were reactive with at least two of the six LD control samples and exhibited higher variance in reactivity against the control samples, with the exception of DbpB6 and DbpA4-B6 which had general non-specific interactions across all samples. Additionally, peptides that showed lower inter-correlation, indicating the ability to capture unique antibody signatures were included in the multiplexed panel. The nine selected peptides with each spotted in duplicate along with three positive and four negative internal controls, constituted a total of 25 immunoreaction spots on our sensing array. This carefully curated multiplexed panel, which includes unique epitope-specific targets, enhances our capability to detect distinct epitope-specific immune signatures that single-plex tests may not detect. The selected peptides used for the training of our final xVFA diagnostic model were modVlsE-FlaB, Var2FlaB, BBA64-7, OspC1, OppA4, p-35 21, ErpP59, Flil-B4, and BBA73 196-99.”

2. The authors used 4 cohorts. It was confusing in the text where some cohorts were referred by number and some by abbreviations. The authors should generate a concise summary figure/table to show the characteristics, number of patients and their usage in the study, e.g. for peptide screening, machine learning training/testing, etc.

Response: The characteristics, patient pool for each category and the description of the sample cohorts are described in detail in the methods section. Based on the reviewer comment, we have revised the manuscript to include consistent cohort number whenever sample testing is referenced throughout the manuscript. In brief, samples from cohort 1 were used in epitope mapping of major *Borrelia* antigens. Samples from cohort 2 were used for peptide screening in the xVFA platform. Samples from cohort 3 were used in training and testing the machine learning based diagnostic model. Further, an additional column has been included in Supplementary Table 3 to specify each sample from this cohort used in either training or testing (validation) of the diagnostic model. Sample cohort 4 from the CDC's Lyme disease repository was used for blinded validation of the developed platform.

The following revisions were made in the methods section under Clinical samples sub-section:

“Sample cohort 1 was used for epitope mapping and identification of *Borrelia burgdorferi* specific targets and consisted of Lyme disease sera...”

“Sample cohort 2 was used for screening of the peptides on the xVFA assay platform and included...”

“Sample cohort 3 was used for training and testing of the machine learning based diagnostic model. The Cohort consisted of...”

“Sample cohort 4 was used for blinded evaluation of the xVFA platform and included ...”

3. Figure 1g, shows large error bar and no statistics.

Response: Figure 1g shows error bars for three replicates for each condition. The figure caption has been revised to include the information. It now reads "... individual immunoreaction spot (N=3)."

4. I think that Figure 2c-e should be Figure 2b-d. Also note capitalisation of figure names is not consistent.

Response: The figure 2c label was missing and was corrected in the revised manuscript. The inconsistencies with figure name capitalization have also been correct in the revised manuscript.

5. The IRB approval document numbers for the three centres should be quoted.

Response: Samples obtained at UCLA from the LDB and CDC biobanks were obtained with de-identified labels and are not considered human subjects research by UCLA's IRB. The samples collected by the LDB were obtained through LDB sponsor protocol approval number Advarra IRB protocol Pro00012408.

Reviewers' comments:

Reviewer #1 (Remarks to the Author):

The authors have addressed all previously raised issues, and the manuscript is suitable for publication.

Reviewer #2 (Remarks to the Author):

The authors have made significant requested revisions that I believe strengthen their manuscript.

Reviewer #3 (Remarks to the Author):

The authors have addressed most of the issues previously noted on the first submission of this manuscript. However, a few additional clarifications and corrections are needed before it should be considered for publication.

1. Lines 101-107: Although the combined detection of IgM and IgG antibodies increased the sensitivity of their single-tier test, it might lead to false positivity in patients with persistent IgM antibodies from a prior infection who are now presenting with a non-Lyme related illness. The authors could add a comment about this possibility.

2. ModVlsE is presented as a selected peptide throughout the text, but Figures 2,3 and Supplemental Fig4 mention “ModC6”. Is this correct?

3. A typographic error is found on Supplementary Table 2. It should read “modVlsE-FlaB” rather than “modClsE-FlaB”.

4. Lines 290-293: I suggest adding a comment indicating that testing outside a central laboratory would need to be contingent upon compliance with existing Clinical Laboratory testing regulatory requirements.

Reviewer #4 (Remarks to the Author):

The authors have made extensive revisions to the manuscript and should be commended for their detailed revisions.

I am surprised by the much less-good performance of the test for the blinded study when compared to the unblinded/training data. Do the authors have suggestions as to why this is?

For the blinded study the low sensitivity and high specificity would imply that the authors can find the negatives with confidence, but conversely cannot find the positives with the same high confidence, ie lots of people with the disease will not show positive. This is not ideal, especially for a device used in disease surveillance - the authors should describe the associated challenges that this presents with community testing.

I also think with such small numbers in the clinical study (and only 8 negatives in the cohort) expressing the results as % is misleading. If the authors do show % then they should provide confidence intervals.

The reference to this being a point of care test remains difficult as the study involves using serum samples, which need a sample processing step that is not integrated in the assay workflow/instrument. I think that if this is the case, this should be made clear. The authors should describe the associated challenges that this presents with community testing and surveillance.

Reviewers' comments:

Reviewer #1 (Remarks to the Author):

The authors have addressed all previously raised issues, and the manuscript is suitable for publication.

Response: We thank the reviewer for the positive remark.

Reviewer #2 (Remarks to the Author):

The authors have made significant requested revisions that I believe strengthen their manuscript.

Response: We thank the reviewer for the positive remark.

Reviewer #3 (Remarks to the Author):

The authors have addressed most of the issues previously noted on the first submission of this manuscript. However, a few additional clarifications and corrections are needed before it should be considered for publication.

Response: We thank the reviewer for the positive remark.

1. Lines 101-107: Although the combined detection of IgM and IgG antibodies increased the sensitivity of their single-tier test, it might lead to false positivity in patients with persistent IgM antibodies from a prior infection who are now presenting with a non-Lyme related illness. The authors could add a comment about this possibility.

Response: We expect IgM antibody levels to decline after acute infection. IgG antibodies may remain at a background level in patients with a prior infection, and this is always a concern with any serology test, which we now discuss in the discussion section of the revised manuscript.

“It is possible that patients with previous LD infections that had recovered may have remaining low levels of IgG that would yield a false positive test. Future guidelines could inform patients or healthcare providers to interpret the test results in the context of the patient’s history and other signs and symptoms.”

2. ModVlsE is presented as a selected peptide throughout the text, but Figures 2,3 and Supplemental Fig4 mention “ModC6”. Is this correct?

Response: We thank the reviewer for raising the concern, we corrected the discrepancy in labelling in the revised manuscript at the respective locations.

3. A typographic error is found on Supplementary Table 2. It should read “modVlsE-FlaB” rather than “modClsE-FlaB”.

Response: The error is corrected in the revised manuscript.

4. Lines 290-293: I suggest adding a comment indicating that testing outside a central laboratory would need to be contingent upon compliance with existing Clinical Laboratory testing regulatory requirements.

Response: The sentence has been amended in the revised manuscript. The statement now reads, “The high specificity of our xVFA combined with the comparable sensitivity in a POC format would be suitable in a rapid, POC test setting such as local pharmacies, clinics, or other POC settings, as per CLIA guidelines and regulations, enabling healthcare providers to quickly...”

Reviewer #4 (Remarks to the Author):

The authors have made extensive revisions to the manuscript and should be commended for their detailed revisions.

Response: We thank the reviewer for the positive remark.

I am surprised by the much less-good performance of the test for the blinded study when compared to the unblinded/training data. Do the authors have suggestions as to why this is?

Response: The CDC samples consisted of completely different sample types and composition compared to the training cohort from the Lyme Disease Biobank. The cohort included early LD samples with 4 acute infections, 4 early LD samples with convalescent infection, 4 late-stage LD samples including 2 Lyme arthritis and 2 Lyme carditis, 12 look-alike disease samples that are known to cross-react with current commercial LD tests and 8 healthy control samples from LD endemic and non-endemic regions. The reviewer may be referring to the performance in a subset of patient samples included in the CDC cohort (4 samples), which likely did not yet develop an antibody response (IgM or IgG), and therefore went undetected using the xVFA and the STTT tests and only one additional sample was detected using the MTTT IgM diagnosis approach. We highlight this aspect in the results section in discussion of the performance of the xVFA in the CDC cohort, “Both the single tier xVFA and the STTT did not detect any of the acute LD samples in the panel. This could be attributed to the low LD-specific antibody levels in this particular cohort of early acute LD samples, requiring follow-up testing until a detectable immune response can be measured using these tests. Only the modified two-tier testing algorithm performed slightly better using the IgM-based diagnosis and was able to classify one additional acute LD sample as LD positive.”

Additionally, it's important to note that we evaluated the performance of the xVFA using the CDC LD repository samples to benchmark the performance of our developed tests against established reference tests on the same patient samples. Figure 4b, 4c and Table 1 present a summary of this evaluation, comparing the performance of the single-tier xVFA test against standard reference tests and two-tier diagnosis outcomes. Due to the distinct sample compositions, the performance of the xVFA in terms of sensitivity and specificity results cannot be directly compared to the initial training results. To illustrate the xVFA's effectiveness on a cohort that mirrors the same balanced sample distribution, we have now validated the diagnostic model with 30 serum samples, which were previously not tested or used in training of the diagnostic model. The samples consisted of 15 confirmed LD positive samples, all of which were positive for IgM antibodies, indicating acute infections with early immune response. Further, it included 15 LD negative confirmed samples from LD endemic regions confirmed negative using two-tier serology. The samples were from the Bay Area Lyme Disease Biobank and were selected to reflect a balanced distribution as in the training cohort to enable a comparison of performance between the training phase and the validation phase. Employing the originally trained diagnostic model, the xVFA attained a

sensitivity of 95.5% and a specificity of 100% with these samples, demonstrating reliable precision and effectiveness. While keeping the CDC cohort results to show concordance with the two-tier tests, we now report sensitivity and specificity in this separate validation study with the cohort of evenly distributed 30 samples.

For the blinded study the low sensitivity and high specificity would imply that the authors can find the negatives with confidence, but conversely cannot find the positives with the same high confidence, ie lots of people with the disease will not show positive. This is not ideal, especially for a device used in disease surveillance - the authors should describe the associated challenges that this presents with community testing.

Response: We would like to emphasize that the relatively lower sensitivity observed for the blinded CDC panel is due to four acute-stage LD infections, which likely had not yet developed an antibody response, and remain largely undetected in both the xVFA and two-tier serology tests (STTT and MTTT) recommended by the CDC.

Furthermore, as we discussed in our revised manuscript:

“A negative result may indicate repeat testing is needed in acute patients with other signs and symptoms after waiting for serum IgM and/or IgG antibody levels to increase further.”

In the context of large-scale testing, a significant proportion of test takers would likely test negative for LD due to the low pretest likelihood of LD in the community. Thus, in such scenarios, it is critical for tests to be highly specific to limit false positive outcomes (which would create a burden on the health-care system) and have a high positive predictive value. We concur that the xVFA test, with its 100% specificity and comparable sensitivity with that of the reference laboratory-grade tests are indicative of superior performance when used for large-scale testing such as community-wide testing.

Further, in our revised manuscript we validate the performance of the xVFA and the optimized diagnostic model using a balanced distribution of 30 new samples. This consisted of 15 LD positive samples all of which were positive for IgM antibodies, indicating acute infections with early immune responses. The xVFA demonstrated a 95.5% sensitivity in this panel indicating reliable sensitivity.

I also think with such small numbers in the clinical study (and only 8 negatives in the cohort) expressing the results as % is misleading. If the authors do show % then they should provide confidence intervals.

Response: Due to the unbalanced distribution of LD and non-LD cases within this cohort, we have chosen not to report sensitivity and specificity percentages in the revised manuscript. Instead, the data is now displayed as the number of positive cases in tabular format, as detailed below and in Table 1 of the revised manuscript. Additionally, we evaluate the xVFA's performance against individual reference tests and the standard two-tier (STTT) and modified two-tier (MTTT) diagnoses, using % agreement and % accuracy as the measure across all different sample groups in this cohort. There was 100% agreement with the STTT and 96.6% agreement with the MTTT. The confidence interval for the xVFA's accuracy percentage is also documented in the revised manuscript. The statement in the results section reads, “The xVFA demonstrated an accuracy of 88% (95% CI: 76.7%-99.2%), correctly identifying 8 of the 12 LD cases and all the non-LD cases.”

Table 1. Clinical samples obtained from the CDC to assess the specificity of the xVFA and the corresponding and ground truth information.

Sample Group	Standard two-tier tests (STTT)			Modified two-tier tests (MTT)				Diagnosis			
	IgM/IgG EIA	IgM WB	IgG WB	Zeus VlsE/pepC1 0 EIA	Zeus WCS EIA	Zeus IgM EIA	Zeus IgG EIA	STTT	MTT T IgM	MTT T IgG	xVFA
Early Lyme-Acute	0/4	1/4	0/4	2/4	1/4	1/4	0/4	0/4	1/4	0/4	0/4
Early Lyme-Convalescent	4/4	4/4	1/4	4/4	4/4	4/4	1/4	4/4	4/4	1/4	4/4
Late Lyme disease	4/4	2/4	4/4	4/4	4/4	4/4	4/4	4/4	4/4	4/4	4/4
Look-alike diseases	2/12	0/12	0/12	1/12	1/12	1/12	0/12	0/12	0/12	0/12	0/12
Healthy control	2/8	0/8	0/8	0/8	3/8	3/8	0/8	0/8	0/8	0/8	0/8

The reference to this being a point of care test remains difficult as the study involves using serum samples, which need a sample processing step that is not integrated in the assay workflow/instrument. I think that if this is the case, this should be made clear. The authors should describe the associated challenges that this presents with community testing and surveillance.

Response: We appreciate the reviewer's observation regarding the use of serum samples, which are indeed central to the xVFA test training and validation. Like other rapid and centralized lab serological tests, the xVFA utilizes serum as a sample type for testing. This preference is due to the serum component providing a clear sample for analysis upon blood coagulation and cell removal, and it is also higher in antibody concentration, which is the target biomarker for LD diagnosis. Additionally, **serum is typically preferred for the long-term storage of well-characterized patient samples and was available from the biobanks utilized in this study.** Due to the complexities associated with LD diagnosis, biobanks serve a very important need for diagnostic technology development and their serum samples with the associated ground truth diagnoses provided are an irreplaceable part of any new LD POC test development effort.

We also highlight that several POC lateral flow tests on the market use serum samples as an acceptable sample format and utilize either a rapid extraction kit or an alternate serum extraction protocol. For instance, the Sofia Lyme 2, an FDA-cleared first-tier rapid Lyme serology assay developed by Quidel Ortho, is a lateral flow test that uses serum or plasma extracted on-site. Despite the additional preprocessing step, there are rapid POC solutions for obtaining serum or plasma samples from a blood draw, thus enabling subsequent measurement of analytes using a rapid test.

However, as pointed out by the reviewer, the xVFA test, as reported in this study, does not include a preprocessing step as part of the workflow. Therefore, we have clarified this concern in our discussion of the POC application for this test. In the revised version of the manuscript attached with this response, we include in the discussion section:

“The xVFA test is optimized for use with serum samples, which are obtained by removing blood cells from patient blood. The isolation of coagulants from the blood aids in analysis and results in

a higher antibody concentration sample that is conducive to the detection of early disease. When deployed for POC testing or testing in resource-limited regions, the xVFA could be coupled with a rapid serum extraction step to generate 20 μ L of serum or plasma that can be utilized for testing on-site.”

REVIEWERS' COMMENTS

Reviewer #4 (Remarks to the Author):

A new cohort are 70 individuals, with associated results from these new samples is included in the revision. This involved a final validation study that used 30 samples (with the other 40 being used as a training dataset).

It appears that the new samples used in the validation were not evaluated as part of a blinded study (the 15 positive and 15 negative were chosen). Can this be clarified in case I missed this. The accuracy remains at 88%. The authors should also provide information as to how the samples were chosen and run.

The sensitivity/specificity should be cited with confidence intervals. Quoting a sensitivity of 95.5% infers a precision of 1/500 for 30 samples.

If the authors propose to use a rapid extraction kit or an alternate serum extraction protocol as part of the testing, they should indicate in the discussion whether the additional burdens of cost and complexity would still enable the widespread application of this test in community screening.

Reviewer #4 (Remarks to the Author):

A new cohort are 70 individuals, with associated results from these new samples is included in the revision. This involved a final validation study that used 30 samples (with the other 40 being used as a training dataset).

It appears that the new samples used in the validation were not evaluated as part of a blinded study (the 15 positive and 15 negative were chosen). Can this be clarified in case I missed this. The accuracy remains at 88%. The authors should also provide information as to how the samples were chosen and run.

The new samples were also blinded. To clarify, our general process was as follows: The UCLA team that is running the xVFA test requested samples from the Lyme Disease Biobank (LDB) that were evenly distributed between LD-positive and LD-negative for the training and validation panels. The LDB then sent the samples to UCLA with unidentified sample labels without ground truth information. Although the sample distribution was predetermined, the ground truth information was only revealed after the completion of the xVFA assay and sending the test data back to the LDB.

In our original submission, we trained the machine learning model using 40 samples (20 LD-positive and 20 LD-negative), with an additional 10 blinded samples for validation of the optimized model. During subsequent review rounds and discussions with the editor, we realized the CDC panel was not representative and not evenly distributed, so we requested 20 more samples from the LDB, forming a separate validation panel of 30 samples (15 LD-positive and 15 LD-negative). As discussed above, the individual labels were only revealed post-assay, upon sharing of our results with the Biobank.

Another key point that reflects robustness and generalizability for the test is that the machine learning model used to evaluate all samples in the CDC and LDB validation panels remained unchanged after the initial training. The model was not exposed to any validation samples beforehand, ensuring that the evaluation was consistent and unbiased across different sample sets.

Regarding accuracy, it was 88% in the CDC cohort and 97.7% in the validation cohort (calculated from the confusion matrix). Details on sample selection and testing is provided in the methods section of the manuscript under Clinical samples.

The sensitivity/specificity should be cited with confidence intervals. Quoting a sensitivity of 95.5% infers a precision of 1/500 for 30 samples.

The confidence interval for sensitivity is updated in the revised manuscript. Sensitivity: 95.5% (95% CI: [89.5%, 100%]).

If the authors propose to use a rapid extraction kit or an alternate serum extraction protocol as part of the testing, they should indicate in the discussion whether the additional burdens of cost and complexity would still enable the widespread application of this test in community screening.

Serum extraction is a routine procedure performed for the detection of biomarkers in blood samples. As highlighted in our previous response to reviewers, the Sofia Lyme 2, an existing point-of-care (POC) platform for just first-tier LD testing, uses a separate serum extraction kit as an integral part of the test. Given the established use of this approach in existing serology tests, we do not anticipate any significant cost burden or other challenge in the serum extraction process or its deployment.